# iMUT-seq: high-resolution DSB-induced mutation profiling reveals prevalent homologous-recombination dependent mutagenesis

Aldo S. Bader [ID] [1,2,3] ✉ & Martin Bushell [1,4] ✉

DNA double-strand breaks (DSBs) are the most mutagenic form of DNA damage, and play a significant role in cancer biology, neurodegeneration and aging. However, studying DSB-induced mutagenesis is limited by our current approaches. Here, we describe iMUT-seq, a technique that profiles DSB-induced mutations at high-sensitivity and single-nucleotide resolution around endogenous DSBs. By depleting or inhibiting 20 DSB-repair factors we define their mutational signatures in detail, revealing insights into the mechanisms of DSB-induced mutagenesis. Notably, we find that homologous-recombination (HR) is more mutagenic than previously thought, inducing prevalent base substitutions and mononucleotide deletions at distance from the break due to DNA-polymerase errors. Simultaneously, HR reduces translocations, suggesting a primary role of HR is specifically the prevention of genomic rearrangements. The results presented here offer fundamental insights into DSB-induced mutagenesis and have significant implications for our understanding of cancer biology and the development of DDR-targeting chemotherapeutics.

DNA double-strand breaks (DSBs) are highly mutagenic lesions that commonly result in genomic base substitutions, deletions and chromosomal rearrangements. DSBs occur in our genomes everyday due to exogenous sources, such as radiation, as well as endogenous sources, such as transcription-replication conflicts. As a result, effective DSB-repair (DSBR) is critical for genome maintenance to prevent the toxic accumulation of DSB-induced mutations, which are commonly associated with diseases, most notably cancer. Defects in DSBR are also commonly associated with cancer-prone syndromes as they promote increased mutagenesis, driving carcinogenesis as well as cancer progression.

There are two major DSBR pathways; non-homologous end-joining (NHEJ) and homologous recombination (HR), which operate via distinct mechanisms to provide comprehensive repair. NHEJ provides a rapid pathway of minimal processing followed by ligation, whereas HR is longer and is limited to S/G2-phase, but provides high fidelity repair by using the sister chromatid as a template[1,2].

Whole genome sequencing (WGS) and exogenous reporter systems are currently used to study DSB-induced mutations. WGS can directly quantify genomic mutations after DNA damage, however it has low sensitivity, requiring high levels of damage, and this damage is randomly acquired across the genome providing no insight into how mutations are introduced relative to the site of damage[3]. Reporter systems for DSB-induced mutations often use CRISPR-Cas9 to induce DSBs at exogenous sequences inserted into the genome[4–7], therefore allowing mutation mapping around these DSBs. These introduced cleavage sites are however exogenous to the genome, limiting their applicability. Additionally, these systems induce high levels of DNA damage to enrich for mutations, leading to mutations per loci of up to 100% due to expression of the enzymes for multiple days, and often

[1]Cancer Research UK Beatson Institute, Glasgow G61 1BD, UK. [2]Cancer Research UK/CI, University of Cambridge, Li Ka Shing Centre, Cambridge CB2 0RE, UK. [3]The Gurdon Institute, University of Cambridge, Biochemistry, Cambridge, UK. [4]Institute of Cancer Sciences, University of Glasgow, Glasgow G61 1QH, UK. ✉e-mail: ab2510@cam.ac.uk; martin.bushell@glasgow.ac.uk

show Cas9 specific mutation signatures[4–7]. These approaches have important implications for CRISPR genome editing applications, but are less applicable to specifically studying DSB-induced mutations. Additionally, these studies have allowed the examination of deletions and insertions, however they rarely study translocations and never analyse base substitutions, which is possibly related to the difficulty in mapping these over background[4–7].

Here, we describe our developed iMUT-seq technique that uses an inducible endonuclease system to introduce DSBs across the genome at defined endogenous loci, followed by targeted next-generation sequencing (NGS) to profile the wide variety of DSB-induced mutations at single-nucleotide resolution and with extremely high sensitivity around these DSBs. We then systematically depleted or inhibited most major DSB repair factors from both the NHEJ and HR pathways, characterising their mutational profiles in unprecedented detail.

## Results

### Overview of the iMUT-seq technique

To provide controlled induction of DSBs at known genomic loci, we employed the damage-induced via AsiSI (DIvA) cell-system. Created by the Legube Lab, DIvA utilises an AsiSI restriction enzyme fused to an oestrogen receptor to provide 4-hydroxytamoxifen (OHT) inducible DSBs with 2nt 'TA' overhangs at the AsiSI recognition sites across the human genome. Specifically, we employed the AID-DIvA system that allows rapid degradation of the AsiSI enzyme via indole-3-acetic acid (IAA) treatment and repair of the DSBs (Fig. 1A–C). Previous research has profiled the AsiSI recognition sites that are reproducibly cut[8] and also characterised subsets of these that are preferentially targeted for either NHEJ or HR repair[9].

We then coupled this system to a multiplexed genomic PCR that amplifies 25 regions: 10 NHEJ-prone, 10 HR-prone and 5 uncut control loci that are distributed across regions of both high and low transcriptional activity (Supplementary Data 1). Next-generation sequencing of this amplicon pool profiles the repaired DSBs at very high depth (Fig. 1D), accurately quantifying individual mutations at rates as low as 1 in 200,000 events. The PCR products are ~250–290 bp in length to allow full sequence coverage using paired-end 150 cycle sequencing (Supplementary Fig. 1a). Excluding the amplification primers, the sequencing extends at least 100 bp either side of the DSBs. These amplicons can then be used in any standard DNA library preparation protocol to create NGS libraries (Supplementary Fig. 1a). In addition, inclusion of all primers in one reaction allows amplification of any locus targeted by the primer pool that became translocated (Fig. 1D). Once sequenced, our analysis protocol profiles the wide variety of mutations using a machine learning optimised alignment that ensures even heavily mutated sequences are successfully mapped

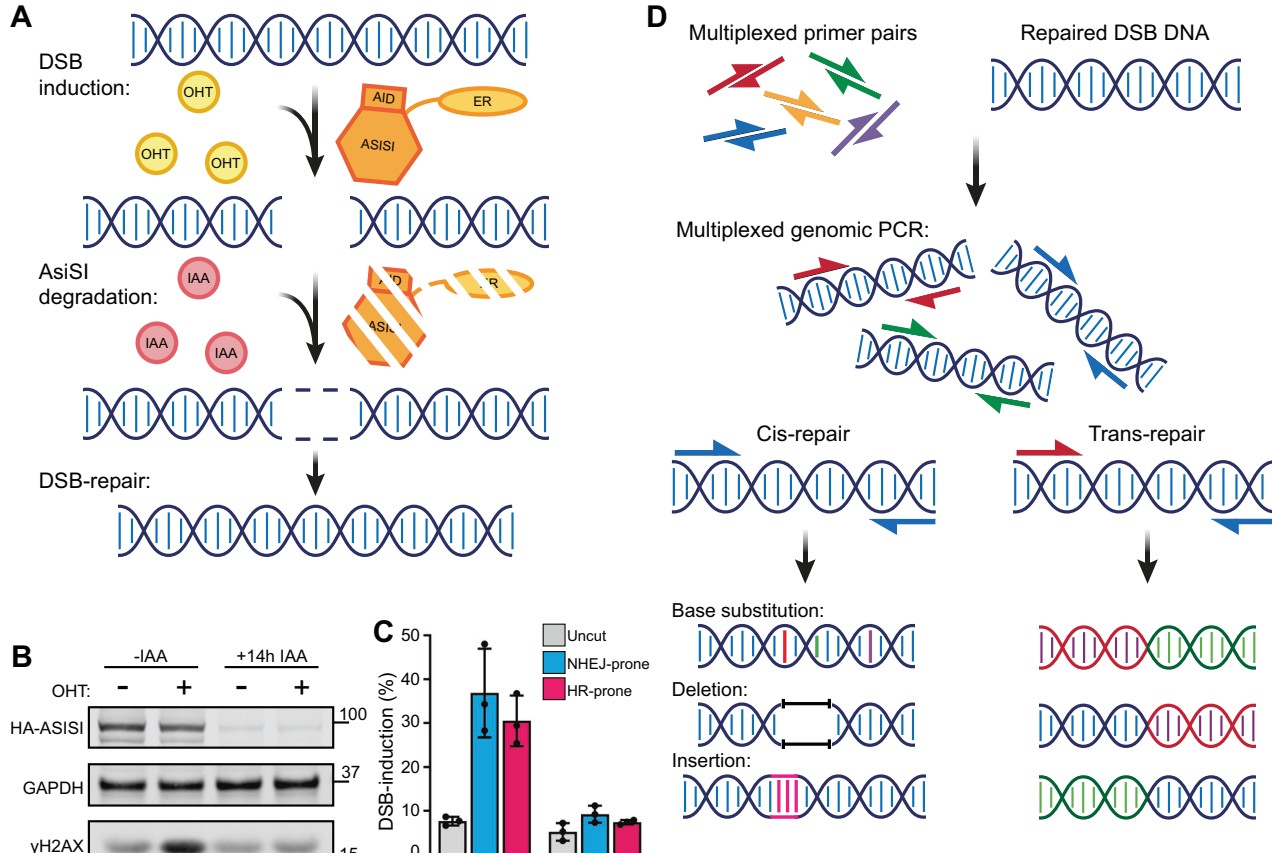

Fig. 1 | **Overview of the iMUT-seq technique. A** Experimental design of iMUT-seq, first Damage-Induced via AsiSI (DIvA) cells are treated with hydroxytamoxifen (OHT) to translocate the AsiSI-ER-AID fusion protein to the nucleus where it generates DSBs. Next, the fusion protein is degraded by activating it's auxin-inducible degron (AID) with auxin (IAA) treatment, allowing the cell to fully repair all the DSBs. **B** Western blot of AID-DIvA cells treated with OHT to induce DSBs, demonstrated by increased γH2AX, and IAA to induce degradation of the AsiSI fusion protein, molecular weight in kDa marked. **C** qPCR quantification of DSB induction at 3 different loci amplified in iMUT-seq; one NHEJ-prone on chromosome 9, one HR-prone on chromosome 17 and one uncut control on chromosome 1, points represent each biological replicate and error bars are S.D., n = 3 independent biological replicates. **D** iMUT-seq pipeline. First, genomic DNA from (**A**) is amplified in a PCR reaction with a multiplex of primers targeting the AsiSI induced DSB sites. This amplifies both DSBs ligated in cis as well as translocated DSBs. This amplified DNA can then be sequenced via NGS, allowing us to profile mutations around DSBs at single-nucleotide resolution, as well as map translocations across the genome. Source data are provided with this paper.

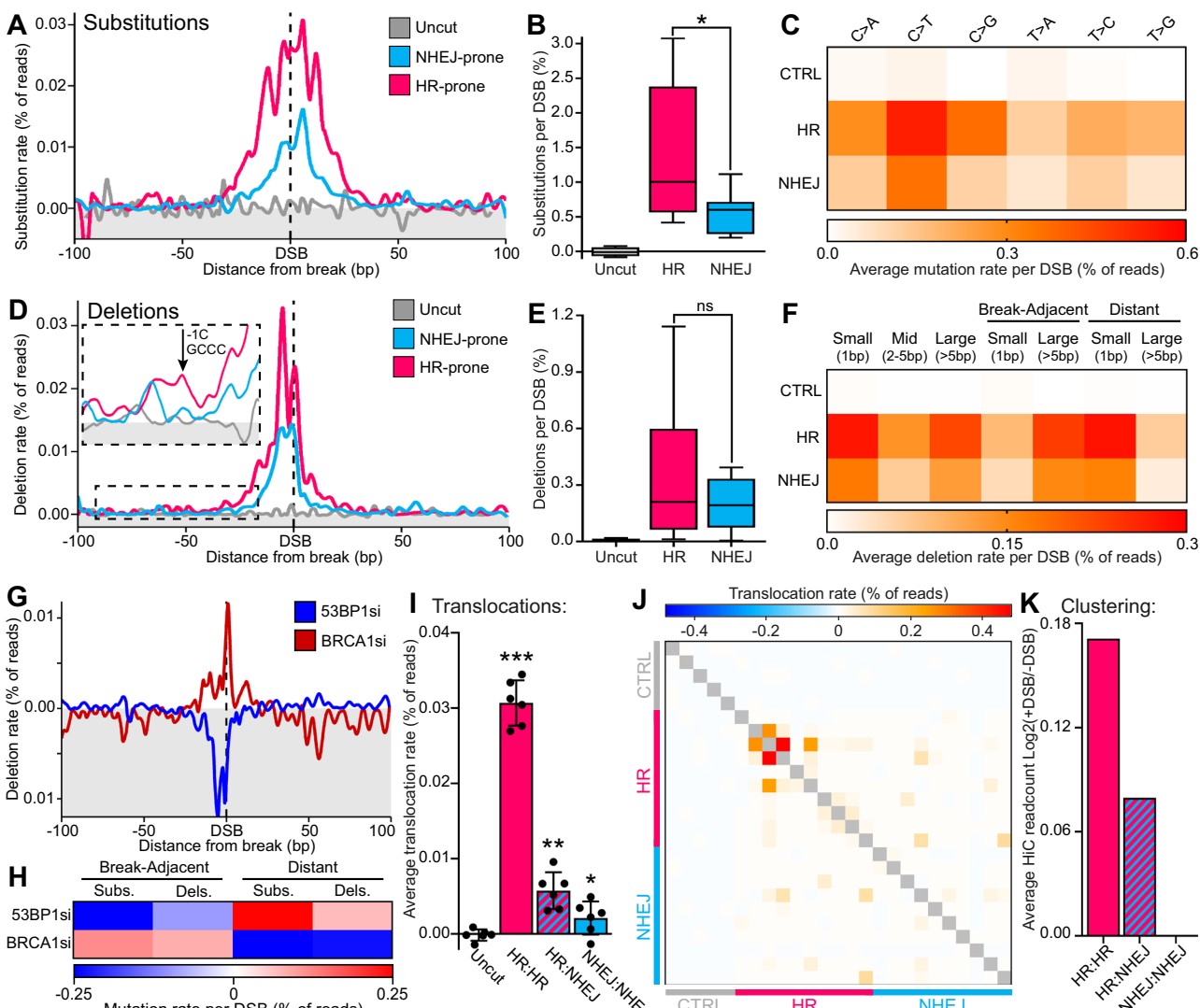

**Fig. 2 | iMUT-seq profiling of HR and NHEJ-dependent DSB induced mutations.** All results show iMUT-seq quantification of DSB-induced mutations at either uncut control, HR-prone or NHEJ-prone loci. **A** Metagene line plot of base substitution rate at either uncut control loci or DSB loci prone to either NHEJ or HR repair. **B** Boxplot of total base substitutions per loci, centre is median, box minima and maxima are interquartile ranges and whiskers are the minimum and maximum values, statistics done via unpaired, two-sided Wilcoxon test, * $p < 0.05$, $n = 10$ loci for HR and NHEJ sites determined from 6 independent biological replicates. **C** Heatmap of base substitution rates per DSB loci. **D** Same as (A) but for deletions, also with a zoomed in section showing distant deletions caused by polymerase slippage at polynucleotide repeats. **E** Same as (B) but for deletions. **F** Heatmap of deletion rates per DSB. **G** Metagene line plot of deletion rate with either 53BP1 or

BRCA1 depletion, relative to control siRNA **H** Heatmap of deletion and substitution rates per DSB that are either adjacent to or distant from the break, with either 53BP1 or BRCA1 siRNA, delta to control siRNA. **I** The average translocation rate between different loci, points represent each biological replicate, and error bars are S.D., all statistics are done relative to the uncut control using a paired, two-sided t-test, * $p < 0.05$, ** $p < 0.01$, *** $p < 0.001$, $n = 6$ independent biological replicates. **J** Heatmap of translocations rates, each row, and each column represent different iMUT-seq loci with each cell being the translocation rate between two loci. **K** Average clustering between the different DSB loci in (**J**), determined by log2 +DSB/-DSB in Capture-HiC[10], showing that DSB clustering is very similar to translocation rates shown in (**I**). Source data and statistics are provided with this paper.

(Supplementary Fig. 1b, c). Sensitivity of iMUT-seq is directly linked to sequencing depth, therefore we sequenced each sample to a target depth of 10 million reads throughout these experiments, though sensitivity could theoretically be increased by sequencing at a higher depth.

## iMUT-seq profiling of HR and NHEJ-dependent DSB induced mutations

We initially compared the mutation profiles of DSB loci that are prone to either NHEJ or HR repair to explore the induction of different types of DSB-induced mutations between these pathways.

We first profiled base substitutions which found a strong peak of mutations that spread to ~20 bp on either side of the DSB (Fig. 2A).

Interestingly, HR-prone loci show a stronger induction of substitutions ($0.56\% - 1.41\%$ per DSB) as well as a wider mutation peak compared to NHEJ-prone loci (Fig. 2A, B), despite previous analysis showing that these loci have similar rates of DSB induction[9,10]. Analysing the base substitution signatures found that C > T mutations are most common, closely followed by C > G and C > A mutations respectively (Fig. 2C). This signature is similar to COSMIC signature 3[11], a well characterised DSB-induced signature found in breast, ovarian and pancreatic cancers.

Quantifying deletions found a similar mutation peak as substitutions (Fig. 2D), but no significant difference between HR and NHEJ-prone loci. (Fig. 2E). Small deletions (1 bp) are most common (0.22% per DSB), closely followed by large deletions ( > 5 bp, 0.17%) and then

mid deletions (2-5 bp, 0.094%) (Fig. 2F). Interestingly, large deletions are almost all adjacent to the break point, whereas small deletions are mostly distant (Fig. 2F, Supplementary Fig. 2A). This is surprising as distant deletions are not generally described at DSBs, but this data indicates they are more common than the well-studied break-adjacent deletions. Further investigation revealed these distant deletions occur at polynucleotide repeats (Fig. 2D), and we hypothesise that these occur due to polymerase slippage at resected overhangs[12–16], which we explore further later. Very few DSB-induced insertions were identified (0.05% per DSB), suggesting they occur infrequently at DSBs (Supplementary Fig. 2B, C), in contrast to previous reports using CRISPR-induced DSBs which found high levels of insertions, especially 1-2bp[5,6,17]. This discrepancy has been well documented at CRISPR cut sites[17–22], and is considered a CRISPR-specific mechanism possibly due to the Cas9 protein remaining bound at the break[18,19,23]. Of interest, others have previously shown using restriction enzyme-based reporters that insertion rates are low even under highly mutagenic treatments[4].

To further investigate the potential NHEJ and HR-specific mutations at DSBs, we conducted depletion of the NHEJ and HR factors 53BP1 and BRCA1, respectively. 53BP1 depletion reduced break adjacent deletions (−0.10%) and promoted distant deletions relative to control siRNA (+0.07%), whereas BRCA1 depletion had the opposite effect; increasing break adjacent deletions (+0.08%) and reducing distant deletions (−0.24%) (Fig. 2G, H). In addition, we see the same impact on base substitutions (Fig. 2H). This suggests that the break-adjacent mutations we observe are mostly NHEJ-dependent, likely as a result of end-processing. Whereas the distant mutations are mostly HR-dependent, which may be due to polymerase error and slippage during re-polymerisation of the resected DNA during HR, explored later.

A common cause of deletions at DSBs is via microhomology-mediated end-joining (MMEJ). This utilises short stretches of homology, created by digestion of the broken DNA ends, to facilitate ligation[24–26]. We are therefore able to quantify these microhomologies, finding they occur infrequently compared to other deletions (0.068% per DSB) (Supplementary Fig. 2d, e), but do represent larger than average deletions of up to ~70 bp (Supplementary Fig. 2f), with relatively short homologous stretches of only 2–7 bp (Supplementary Fig. 2g)[4,6,25,26].

Mapping translocation events found that HR-prone loci were specifically susceptible to translocations (0.031%) whereas NHEJ-prone loci rarely translocated at all (0.002%) (Fig. 2I), with a further interrogation of this yielding an interesting bias towards a small number of HR-prone loci (Fig. 2J). Previous reports have shown that certain DSB loci cluster together within the nucleus, which has been suggested as a source of translocations[10]. Analysis of the clustering of these DSBs found that translocation rates were very comparable to DSB clustering (Fig. 2K), further supporting that translocation at DSBs is driven by this mechanism[10].

This method has allowed us to quantify the rates of the various mutations at DSBs for the first time. On average, we found DSBs to have a mutation rate per site of ~1% for base substitutions, ~0.27% for deletions, ~0.05% for insertions and ~0.01% for translocations (Fig. 2B, E, Supplementary Fig. 2c, h). Although these numbers are likely an underrepresentation due to the frequency of AsiSI cutting being only ~25-30% per locus[10,27,28], it is also not directly quantitative due to the use of bulk cell culture and PCR amplification. As a result, absolute mutation rates cannot be determined and these results should be considered in relative terms between different conditions and mutations types. We have also demonstrated that both NHEJ and HR are significant contributors to DSB induced mutations, contrary to the accepted understanding that NHEJ is error-prone whereas HR is high-fidelity[1,2]. Specifically, these initial results suggest that NHEJ induces break-adjacent deletions (Fig. 2G, H), which are more

commonly large deletions (>5 bp) (Fig. 2f), whereas HR is prone to distant mutations (Fig. 2G, H), such as small deletions (1 bp) (Fig. 2F) and base substitutions. Although, these mutational signatures need further characterisation to be verified and understood.

## Disrupting late stage HR or NHEJ factors results in increased mutation rates compared to initiating factors

To fully investigate the mutagenic mechanism of NHEJ and HR repair, we conducted siRNA mediated depletion of the following NHEJ factors: KU70, Artemis, 53BP1, Pol-λ, XRCC4 and LIG4, resection/HR factors: MRE11, BRCA1, BLM, EXO1, BRCA2, FANCA, RAD51, Pol-δ and Pol-ε and the SSA factor RAD52 (Supplementary Data 2) (Supplementary Fig. 3a), as well as the chemical inhibition of DNA-PK and ATM/ATR.

An overall analysis of the impact of these treatments found that there are distinct trends in DSB-induced mutations as you interfere with different stages of both NHEJ and HR. Depletion of early repair pathway components resulted in a reduction of all types of mutations relative to control siRNA (−1.5% per DSB), whereas depletion of components further down the repair pathways resulted in a progressive increase in mutagenesis (up to +3%) (Fig. 3A). This trend also applied proportionally across the break with mutations spreading further from the break as they increased in rate (Fig. 3B, C, Supplementary Fig. 3B, C). Interestingly, translocations followed this trend for NHEJ depletions, but all HR factor depletions significantly increased translocation rates (+0.5-2 fold) (Fig. 3D). There are some notable exceptions to this trend, such as DNA-PKi which caused a remarkable increase in base substitutions and large deletions, likely due to the specific mechanism of this inhibition[29].

## NHEJ induces mutations around DSBs but protects against large-scale loss of genome integrity

The early NHEJ factors KU70, Artemis, 53BP1 and PNKP all showed a similar mutation profile in our initial analysis (Fig. 3). Further investigation of these factors showed a clear trend as they all significantly reduced base substitutions (−0.4 – −0.6 fold relative to control siRNA per DSB) and deletions (−0.25 – −0.6 fold) around the break, and all have very comparable profiles (Fig. 4A, B). Depletion of Artemis, 53BP1 and PNKP all reduced medium and especially large, break-adjacent deletions (−0.25 – −0.5 fold), but caused an increase in small distant deletions (+0.1–1.3 fold) (Fig. 4C), consistent with a switch from NHEJ to HR mutations signatures (Fig. 2). Interestingly, KU70 depletion also results in a decrease in distant small deletions (−0.8 fold).

Treatment with an inhibitor of the DNA-PK kinase yielded remarkably different results from these siRNA mediated depletions, resulting in a strong induction of base substitutions and deletions specifically at the break point, with a majority of these being large deletions (+5.8 fold) (Supplementary Fig. 4a, b). This mutation signature greatly supports the mechanism of DNA-PKi locking the complex onto DNA ends[29], leading to the endonucleolytic cleavage of the DNA-PK bound DNA[30,31], as this would specifically produce large deletions at the break site.

Quantifying translocations gave similar results, with all depletions reducing or not significantly altering translocation rates (−0.1 – −0.8 fold) (Fig. 4D), except for DNA-PKi which results in an increase (+1.5 fold) (Fig. 4D, Supplementary Fig. 4c). A closer look at the translocation maps revealed that with 53BP1 and PNKP depletions some events decrease in frequency while others increase (Fig. 4E, Supplementary Fig. 4d), suggesting early NHEJ could have a role in suppressing translocations at certain genomic loci.

Reduced mutations with NHEJ depletion does support the notion of NHEJ being a mutagenic repair process[32,33]; however, NHEJ is still thought to preserve genome stability overall[34,35]. To gain further insight into this, we conducted metaphase spreads with etoposide in HCT116 cells as an orthogonal validation. This showed that KU70 depletion significantly increases chromosomal aberrations (Fig. 4F−G),

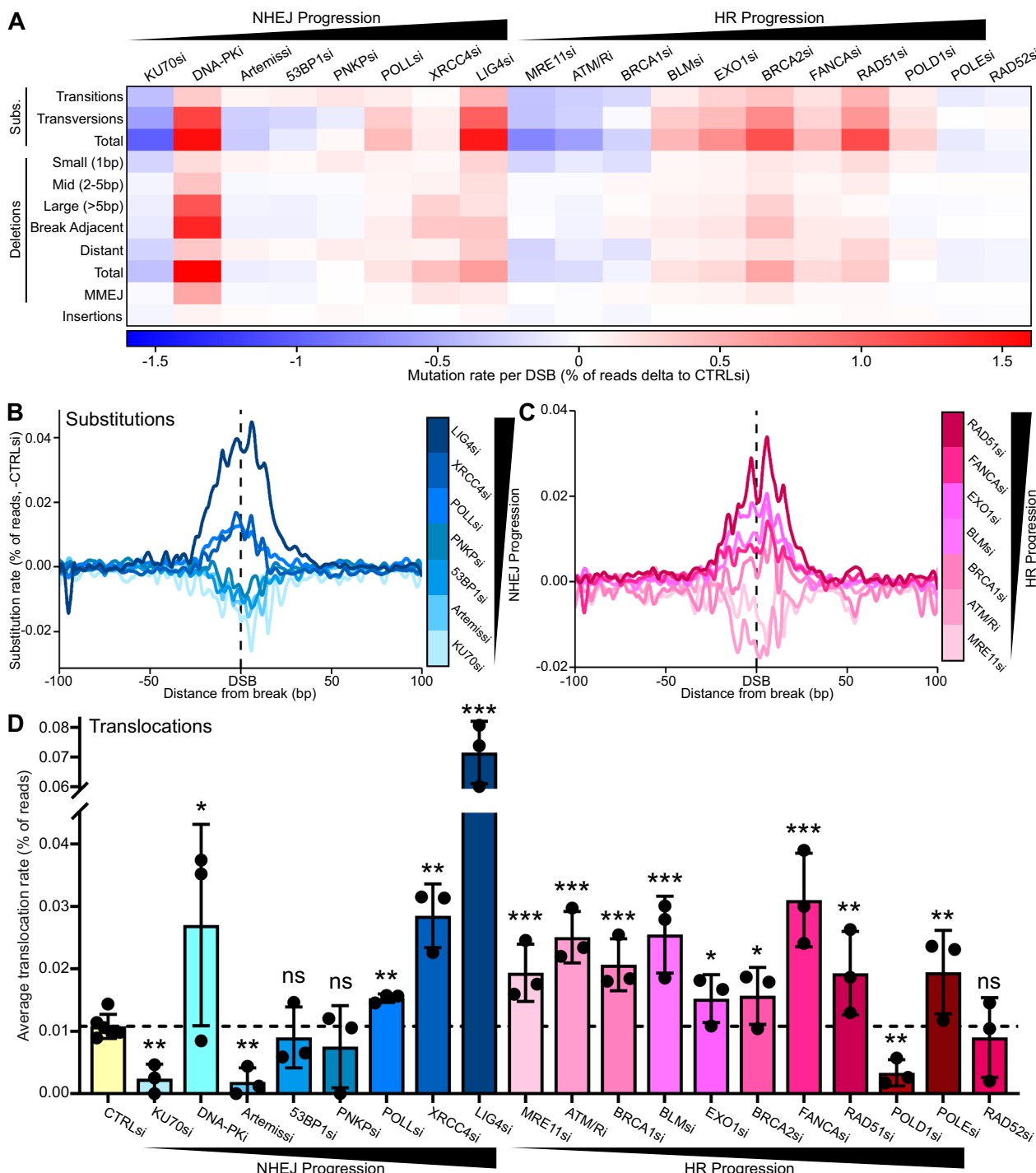

**Fig. 3 | Interrupting DSB repair causes increasing mutation rates the later the interruption occurs in the pathway. A** Heatmap of the average rate of different mutation types per DSB locus, delta to -DSB then delta to control siRNA, quantified by iMUT-seq. **B** Metagene line plots of base substitution rate delta to -DSB then delta to control siRNA, 100 bp either side of AsiSI induced DSBs upon depletion of several different NHEJ repair factors, quantified by iMUT-seq. Legend (right) shows the colours for each siRNA target, as well as a bar depicting the position of the factors in the progression of NHEJ repair. **C** Same as (**B**) but with the depletion of

several HR repair factors. **D** Average translocation rate between DSBs quantified by iMUT-seq upon depletion of 19 different DSB repair factors from both NHEJ and HR repair, points represent each biological replicate and error bars are S.D., all statistics are done relative to the control siRNA result using a paired, two-sided t-test, * $p < 0.05$, ** $p < 0.01$, *** $p < 0.001$, $n = 6$ independent biological replicates for CTRLsi and 3 for all other conditions. Source data and statistics are provided with this paper.

contrary to our previous result (Fig. 4D), and also results in a significant loss of chromosomes (Fig. 4H). This suggests that loss of KU70 does cause a substantial loss of genome integrity following DNA damage, which manifests as deletions larger than can be detected via iMUT-seq, as well as genomic rearrangements and even whole chromosome loss,

likely due to mis-segregations and loss of DNA fragments through mitosis.

Collectively, these results highlight a key role for NHEJ not only in maintaining genome integrity, which has been shown before[34,35], but also in the suppression of HR induced distant mutations. This trade-off

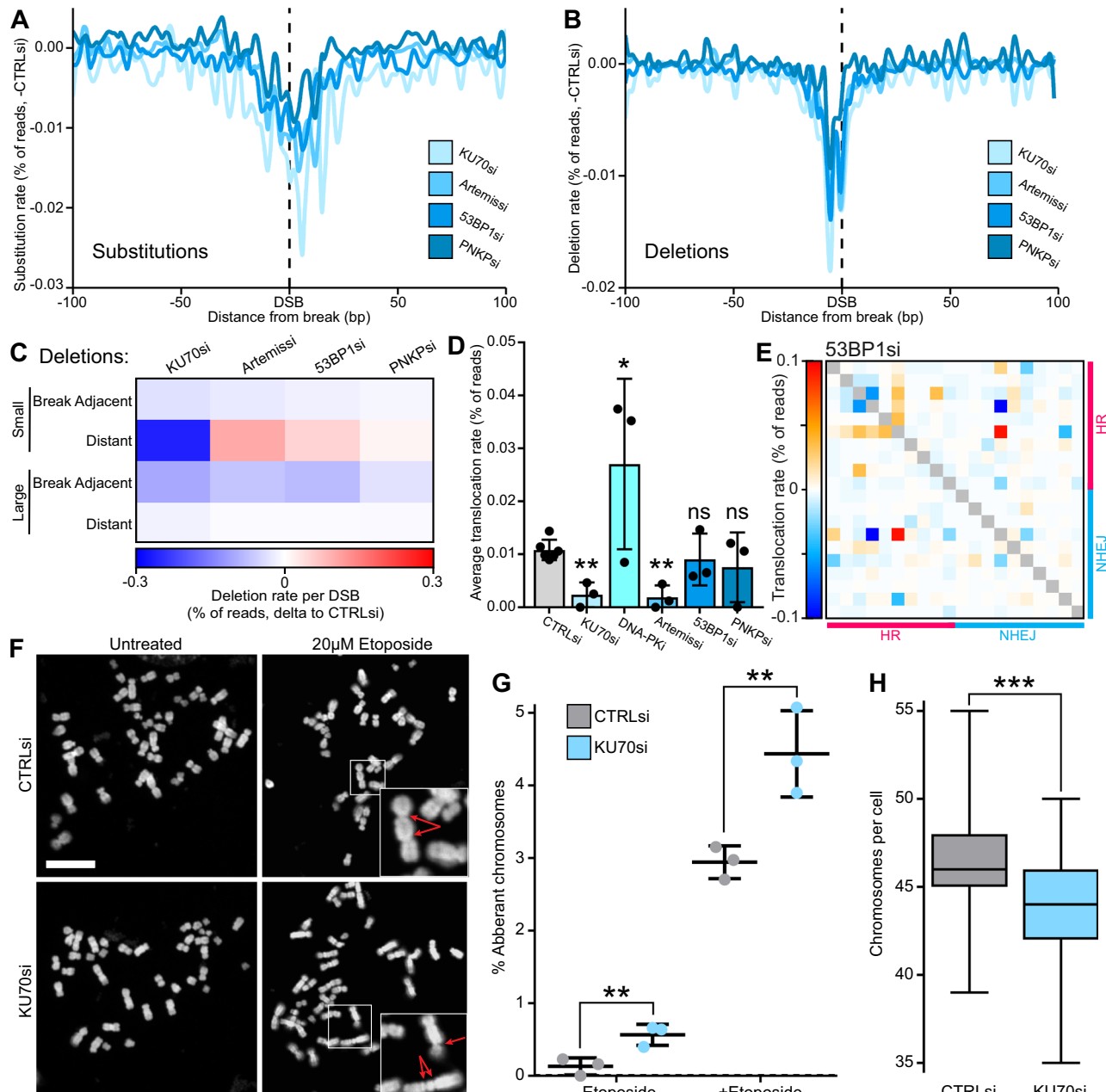

**Fig. 4 | NHEJ induces mutations around DSBs but protects against large-scale loss of genome integrity. A** Metagene line plot of base substitution rate delta to -DSB then delta to control siRNA, 100 bp either side of AsiSI induced DSBs upon depletion of several different NHEJ repair factors, quantified by iMUT-seq. **B** Same as (A) but for deletion rate. **C** Heatmap of the average deletion rates per DSB delta to -DSB then delta to control siRNA. **D** Average translocation rate between DSBs delta to -DSB, points represent each biological replicate and error bars are S.D., all statistics are done relative to the control siRNA using a paired, two-sided t-test, * $p < 0.05$, ** $p < 0.01$, $n = 6$ independent biological replicates for CTRLsi and 3 for all other conditions. **E** Heatmap of translocation rates between DSBs with 53BP1 depletion, each row and each column represent different loci that are either HR or NHEJ-prone with each cell being the translocation rate as between the two loci,

delta to control siRNA. **F** Representative images of metaphase spreads in HCT116 cells with or without 1 hr 20 μM etoposide treatment and with either control or KU70 siRNA, scalebar is 5 μm. **G** Quantification of chromosomal aberrations from (**F**) as a percentage of total chromosomes, points represent biological replicates, centre is mean of replicates and error bars are S.D., statistics done relative to control siRNA using a paired, two-sided t-test, ** $p < 0.01$, $n = 3$ independent biological replicates. **H** Quantification of chromosome number per cell from (**F**) centre is median, box minima and maxima are interquartile ranges and whiskers are the minimum and maximum values,, statistics done using an unpaired, two-sided Wilcoxon test, *** $p < 0.001$, $n = 60$ cells across 3 independent biological replicates. Source data and statistics are provided with this paper.

between NHEJ and HR mutational signatures could be critical in pathway choice at different genomic regions[6,9,36].

## Disruption of late NHEJ processes promotes MMEJ deletions and translocations

In our initial analysis, depletion of the late NHEJ factors POLL, XRCC4 and LIG4 showed a significant deviation from the early NHEJ

phenotype (Fig. 3A–D). An in-depth look at their mutation profiles showed that whereas all three depletions resulted in increased substitutions (+0.5-2.2 fold) and deletions (+0.7-2.1 fold), they showed significantly different mutation profiles (Fig. 5A–C). Further interrogation of the POLL base substitution profile found an interesting feature of increased distant substitutions specifically at HR-prone loci (+1.0 fold) (Supplementary Fig. 5a, b), implicating POLL in an NHEJ to

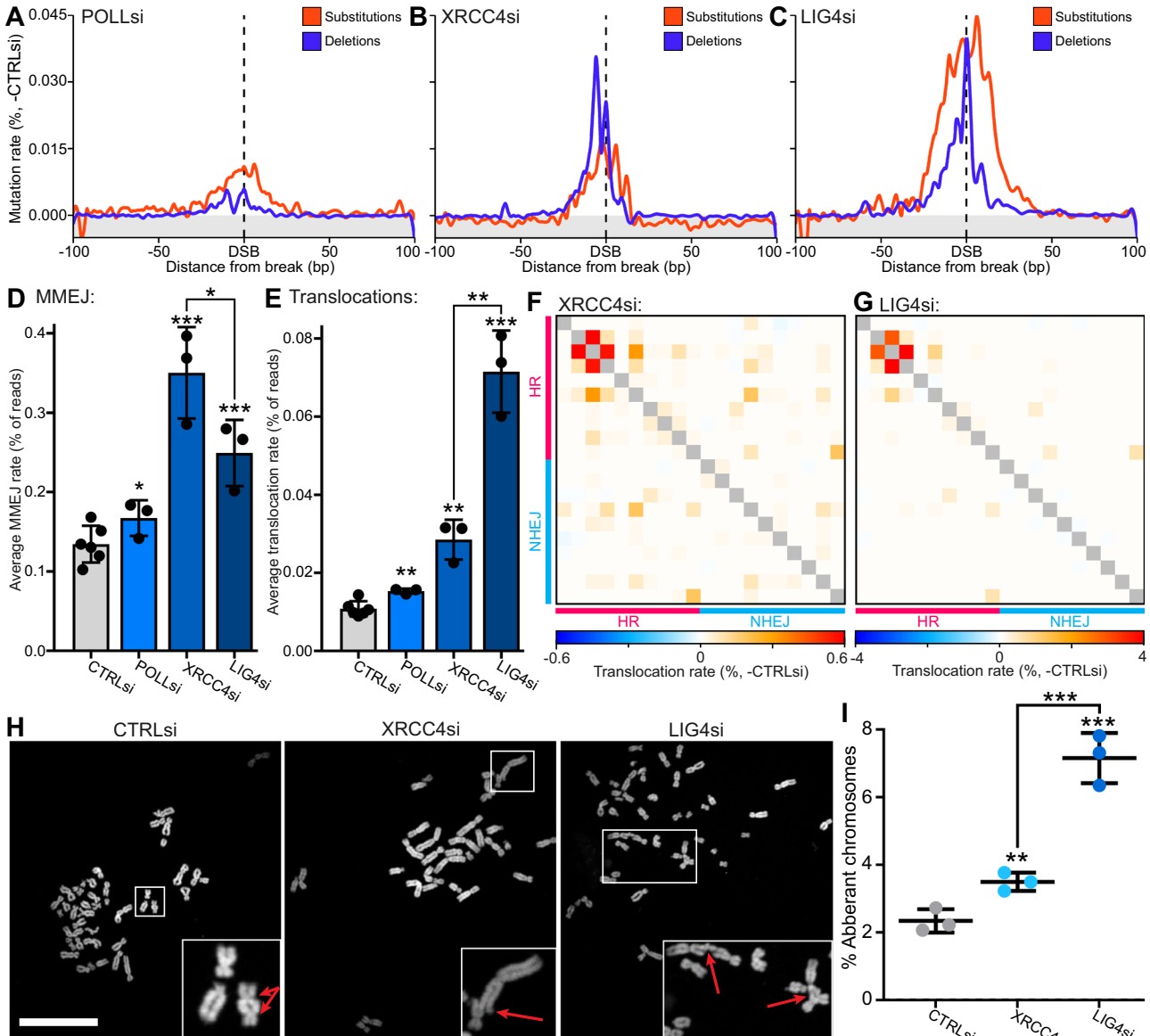

**Fig. 5 | Disruption of late NHEJ processes promotes MMEJ deletions and translocations. A** Metagene line plot of base substitution and deletion rates delta to -DSB and then delta to control siRNA, quantified by iMUT-seq. **B** Same as (**A**) but for XRCC4 siRNA. **C** Same as (**A**) but for LIG4 siRNA. **D** Average rate of microhomology-mediated end-joining (MMEJ) per DSB loci, points represent each biological replicate and error bars are S.D., all statistics are done relative to the control siRNA result using a paired, two-sided t-test, * $p < 0.05$, ** $p < 0.01$, *** $p < 0.001$, $n = 6$ independent biological replicates for CTRLsi and 3 for all other conditions. **E** Same as (**D**) but for translocations per DSB locus. **F** Heatmap of translocation quantified by iMUT-seq with XRCC4 depletion, each row and each column represent different iMUT-seq amplicons of either HR-prone or NHEJ prone loci with each cell being the translocation rate of the two loci, delta to control siRNA. **G** Same as (**F**) but for LIG4 siRNA. **H** Representative images of metaphase spreads with 2 Gy irradiation and with either control, XRCC4 or LIG4 siRNA, scalebar is 10 μm. **I** Quantification of chromosomal aberrations in the metaphase spreads from (**H**) as a percentage of total chromosomes, points represent each biological replicate, centre is mean of replicates, and error bars are S.D., statistics done relative to control siRNA using a paired, two-sided t-test, ** $p < 0.01$, *** $p < 0.001$, $n = 3$ independent biological replicates. Source data and statistics are provided with this paper.

HR switch via gap-filling of resected breaks which has been suggested by previous publications[37,38].

One major distinction in the mutation signatures of these depletions is that whereas LIG4 depletion increased deletions of all lengths, XRCC4 and POLL depletion specifically increased large deletions (+0.3/1.7 fold respectively), with XRCC4 having the strongest effect (Supplementary Fig. 5c). Quantifying MMEJ revealed this is due to XRCC4 depletion leading to a considerable increase in MMEJ (+1.6 fold) (Fig. 5D). This disparity in mutagenic mechanism between XRCC4 and LIG4 is particularly interesting given their close mechanistic relationship[39–41].

Examination of translocation rates found that although depletions of all late NHEJ factors resulted in increased translocations (+0.4-5.6 fold), LIG4 depletion leads to a further increase in translocations at all loci compared to POLL and XRCC4 (Fig. 5E, Supplementary Fig. 5d). However, the translocation maps showed similar translocation events for both XRCC4 and LIG4 depletion, despite the substantial disparity in the magnitude of their effects (Fig. 5F, G). To validate these results, we conducted metaphase spreads in both RPE-1 cells treated with 2 Gy irradiation (Fig. 5H, I) and in HCT116 cells with etoposide treatment (Supplementary Fig. 5e, f). These experiments confirmed our iMUT-seq results of significantly increased chromosomal aberrations with both

XRCC4 and LIG4 depletion, with LIG4 displaying a further significant increase over XRCC4 depletion (Fig. 5H, I).

To further investigate this disparity between XRCC4 and LIG4, we conducted an MMEJ assay based on a GFP reporter system[42] (Supplementary Fig. 5g). A key feature of MMEJ repair is the need for resection to allow annealing of the microhomologies. In the context of XRCC4 and LIG4 depletion, this resection could either be the result of resection during NHEJ via the nuclease Artemis or via HR initiation by MRE11. Individual depletion of MRE11 showed a decrease in MMEJ (−0.3 fold) while depletion of XRCC4 resulted in a significant increase (+0.9 fold) (Supplementary Fig. 5h). Importantly, double depletion of both MRE11 and XRCC4 reduced MMEJ levels down to the level of MRE11 individual depletion (Supplementary Fig. 5h), indicating that XRCC4 siRNA depletion-induced MMEJ is facilitated by MRE11-dependent resection. However, LIG4 depletion showed a limited increase in MMEJ (+0.25 fold) and this increase was not entirely MRE11-dependent (Supplementary Fig. 5h), suggesting LIG4 depletion results in more complex mutagenic mechanisms which may explain it's more broad mutagenic profile when compared to XRCC4 depletion (Fig. 5B, C).

Thus, while XRCC4 depletion promotes MRE11-dependent MMEJ, LIG4 depletion strongly promotes translocation between DSBs, presenting an interesting delineation in the mutation signatures of these cooperating factors. XRCC4 filaments have been implicated in recruiting polymerases to fill-in overhangs at DSBs, which would limit MMEJ at MRE11 resected loci[38,43]. Taken together, this suggests that in the absence of XRCC4, resected overhangs are retained at the DSB, but canonical NHEJ cannot progress, resulting in MMEJ mediated repair. However, depletion of LIG4 would retain blunt ends instead of overhangs, possibly promoting translocations through alternative ligation mechanisms.

## HR repair induces deletions and base substitutions to prevent translocations

In studying the role of early resection factors, we utilised MRE11 and BRCA1 depletions as well as combined inhibition of ATM (10 μM KU55933) and ATR (10 μM VE-821) due to their redundancy[44].

Analysis showed a decrease in both base substitutions (−0.3–0.6 fold) and deletions (−0.2–0.5 fold), particularly at distance from the break (Fig. 6A–C). Interestingly, both MRE11 and BRCA1 depletions not only decreased distant mononucleotide deletions (−0.4–0.9 fold), but also reduced the length of break-adjacent deletions (Fig. 6A, Supplementary Fig. 6A–C). In combination with our previous analysis of HR and NHEJ-prone loci (Fig. 2), this suggests that DSB resection results in increased distant mutations, likely through polymerase error and slippage on resected DNA, but also promotes longer deletions at the break, possibly due to degradation of the exposed single-stranded DNA ends.

To investigate this hypothesis of polymerase-dependent HR-induced mutations, we depleted the HR polymerases Pol-δ (via the POLD1 subunit) and Pol-ε (POLE). These depletions did not cause a peak of mutations at the break like POLL depletion, instead they only affected distant mutations (Fig. 6D, E). We also observed that the prominent C > T and T > C substitutions at DSBs are POLE dependent, while the less common T > A and T > G mutations were Pol-δ dependent (Fig. 6F). Furthermore, depletion of POLE also significantly decreased distant mononucleotide (−0.7 fold) deletions while POLD1 depletion caused them to increase (+0.3 fold) (Fig. 6E), also implicating Pol-ε in the observed slippage events. This data strongly suggests that the distant mutations we observe at DSBs are specific to HR, with Pol-δ showing generally higher fidelity than Pol-ε at DSBs.

To validate these findings, we used the traffic light reporter (TLR) assay for homologous recombination[45]. TLR utilises an I-SceI endonuclease induced DSB at an inactive mutant eGFP gene. This mutant eGFP can be corrected to active GFP when repaired by HR using an exogenous template (Supplementary Fig. 6d). By sequencing over this

region as we did for iMUT-seq, reads that had been successfully repaired by HR could be identified by the corrected eGFP sequence, allowing us to specifically quantify mutations at homologous-recombination repaired loci (Supplementary Fig. 6d, e). Quantifying base substitutions around the reporter DSB revealed that HR-positive reads had prevalent substitutions up to the 100 bp profiled around the break whereas HR-negative reads had no significant induction of these distant substitutions (Supplementary Fig. 6f, g). There was also a clear mononucleotide deletion signature around this DSB, again specifically in HR-positive reads, and the nucleotide context of these deletions again clearly implicates replication slippage as the primary mutagenic mechanism (Supplementary Fig. 6h).

Turning back to the analysis of early resection factors, examination of translocation rates sheds further light on the mutagenic mechanisms of HR, as all treatments significantly increased translocations (+0.8-1.3 fold) (Fig. 6G) at a majority of loci (Fig. 6H–J). This suggests HR has a major role in globally reducing translocations, with BRCA1 depletion even promoting translocations between sites that rarely translocate under control conditions (e.g. those that are NHEJ-prone) (Fig. 6G). This result is consistent with previous work that has observed increased chromosomal rearrangements in homologous recombination deficient cells[46–49].

These results question the role of HR in preventing nucleotide level mutations. Instead, this suggests that HR promotes these mutations due to polymerase errors and slippage when re-polymerising resected DNA, but that HR specifically protects against genomic rearrangements.

## Differing roles of BRCA2, FANCA and RAD52 in preventing DSB-induced mutations

Finally, we investigated the late-HR process of RAD51 loading. Depletion of RAD51 resulted in a broad base substitution peak (+1.4 fold), as well as a precise spike of small break-adjacent and distant deletions (+1.9 fold) (Fig. 7A, D, Supplementary Fig. 7a). Depletion of FANCA and BRCA2, which load RAD51, also induced a broad substitution peak (+0.5-1.4 fold) (Fig. 7B, C) and primarily promoted mononucleotide deletions (+1.0-2.3 fold) (Fig. 7D, Supplementary Fig. 7b, c), similar to the RAD51 profile. In addition, BRCA2 depletion surprisingly resulted in a substantial increase in large break-adjacent deletions (+1.7 fold) (Fig. 7D, Supplementary Fig. 7c). RAD52 depletion showed only a subtle reduction in distant small (−0.25 fold) and break-adjacent large deletions (−0.1 fold) (Supplementary Fig. 7d, e, Fig. 7D). Together, this suggests that both BRCA2 and FANCA contribute to RAD51 loading whereas RAD52, as expected, is dispensable for this. Additionally, BRCA2 may have extra roles that leads to increased large deletions.

To further explore the contributions of BRCA2 and FANCA to RAD51 loading, we conducted immunofluorescence in U2OS cells following etoposide treatment, and stained for RAD51 and RPA70 as a marker of resected DNA. Both BRCA2 and FANCA depletion resulted in significantly reduced RAD51 foci per cell, indicating a significant role in promoting RAD51 recruitment (Fig. 7E, F). Remarkably, whereas FANCA depletion had no effect on RPA70 focus formation, BRCA2 depletion significantly reduced RPA70 foci per cell (Fig. 7E, G). This suggests that BRCA2 also promotes or maintains resection at DSBs, indicating a distinct role earlier in DSB repair. Although this immunofluorescence data is limited and could also be explained by altered resection kinetics or alternative repair mechanisms, it could be indicative of the additional large deletion phenotype we observed (Fig. 7D, Supplementary Fig. 7c).

Surprisingly, translocation mapping tells a different story. Whereas both RAD51 and BRCA2 depletions cause a significant increase in translocations between HR-prone loci (+0.9/0.4 fold respectively), FANCA depletion results in a greater increase (+2.4 fold) (Fig. 7H, I). This indicates an alternative mechanism for FANCA in the prevention of translocations during DSB repair. Interestingly, RAD52

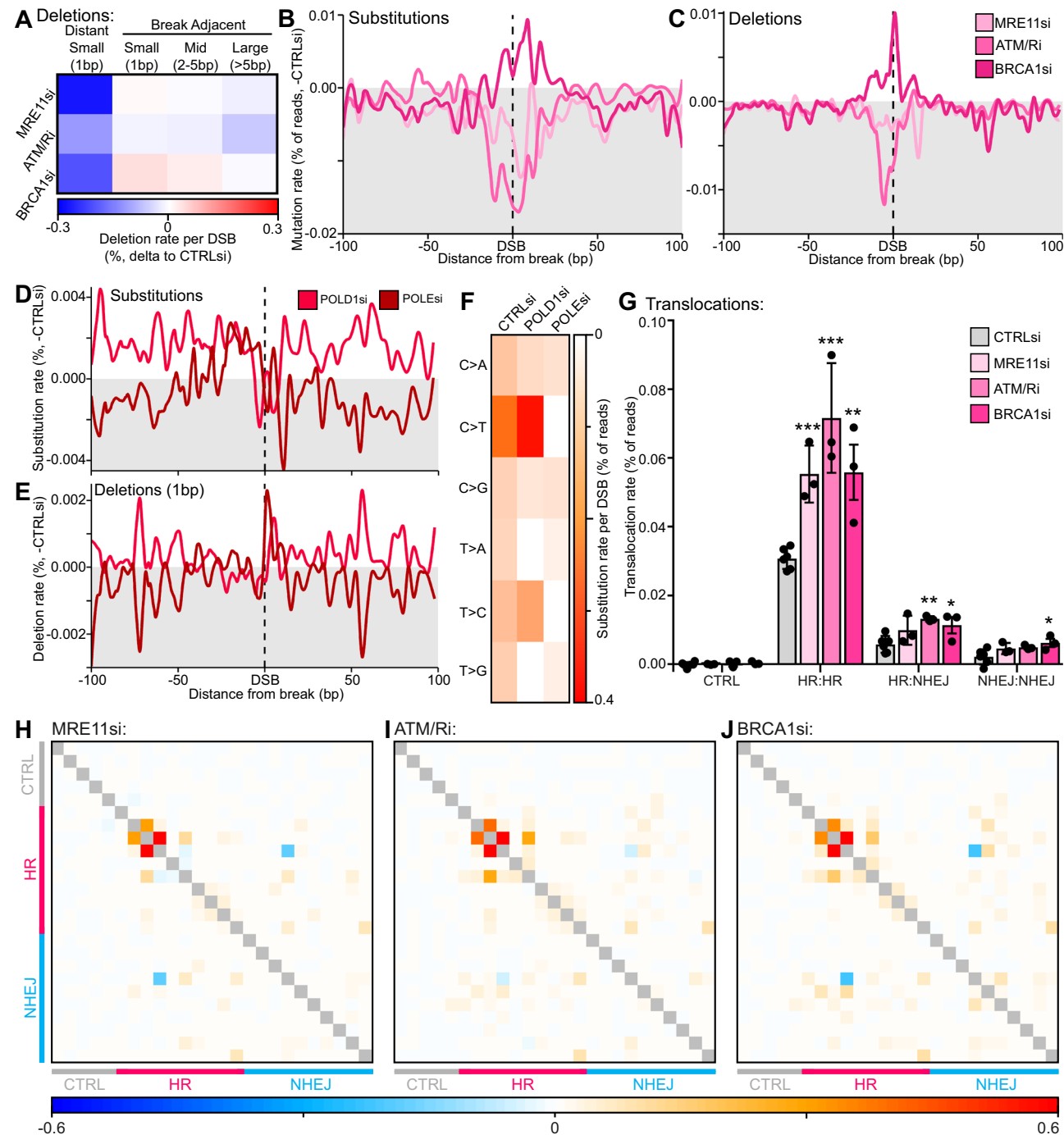

**Fig. 6 | Homologous recombination repair is mutagenic through polymerase error at resected DNA. A** Heatmap of deletion rates per DSB, delta to -DSB then delta to control siRNA. **B** Metagene line plot of base substitution rate delta to -DSB then delta to control siRNA/mock treatment upon depletion or inhibition of MRE11, ATM/ATR or BRCA1, quantified by iMUT-seq. **C** Same as (**B**) but for deletion rate. **D** Metagene line plot of base substitution rate delta to -DSB then delta to control siRNA for POLD1 and POLE depletions. **E** Same as (**D**) but for small, 1 bp, deletions. **F** Heatmap of base substitution rates per DSB loci. **G** Average translocation rate between DSBs at either uncut control, HR-prone or NHEJ-prone loci, with depletion or inhibition of MRE11, ATM/ATR, or BRCA1, points represent each biological replicate, and error bars are S.D., all statistics done relative to control siRNA using paired, two-sided t-tests, * $p < 0.05$, ** $p < 0.01$, *** $p < 0.001$, $n = 6$ independent biological replicates for CTRLsi and 3 for all other conditions. **H** Heatmap of translocation rates with MRE11 siRNA treatment, each row and each column represent different loci that are either uncut control, HR-prone, or NHEJ-prone with each cell being the translocation rate two loci, delta to control siRNA. **I** Same as (**H**) but with ATM/ATR inhibition. **J** Same as (**H**) but with BRCA1 depletion. Source data and statistics are provided with this paper.

depletion causes a reduction in translocations between HR-prone loci (−0.4 fold) (Fig. 7H, I), which in combination with the reduction in small distant and large break-adjacent deletions supports the role of mammalian RAD52 in the mutagenic single-strand annealing (SSA) pathway[50,51].

These results present further evidence that techniques such as iMUT-seq with advanced sensitivity can yield additional insights, even in relatively well-studied areas. The mutational signature of BRCA2 is particularly interesting, because although it resembles a RAD51 signature it also presents a very specific large-deletion signature that isn't

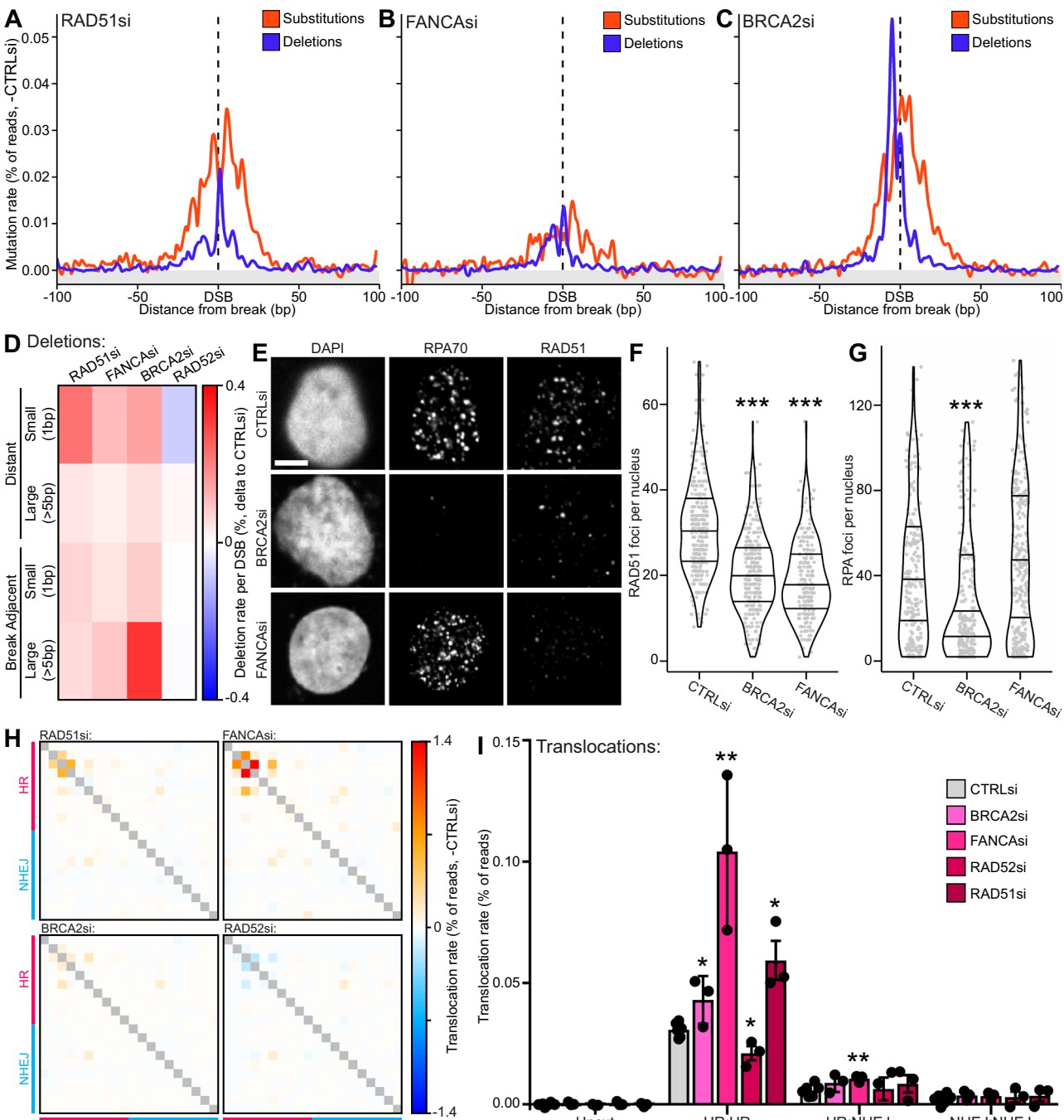

**Fig. 7 | Differing roles of BRCA2, FANCA and RAD52 in preventing DSB-induced mutations. A** Metagene line plot of base substitution and deletion rates delta to -DSB and then delta to control siRNA. **B** Same as (A) but for FANCA siRNA. **C** Same as (A) but for BRCA2 siRNA. **D** Heatmap of deletion rates per DSB loci of different deletion types, delta to -DSB then delta to control siRNA. **E** Immunofluorescence of U2OS cells treated with 5 μM etoposide for 1 hr followed by a 3 hr recovery period, probing for RPA70 and RAD51 with either control, BRCA2 or FANCA siRNA treatment, scalebar is 5 μm. **F** Quantification of RAD51 foci per nucleus in immunofluorescence from (E), lines represent median and interquartile ranges, statistics done using an unpaired, two-sided Wilcoxon test, *** $p < 0.001$, $n = 296$, 307 and 228 nuclei respectively across 3 independent biological replicates. **G** Same as (F) but for

RPA70 foci per nucleus, $n = 251$, 309 and 326 nuclei respectively across 3 independent biological replicates. **H** Heatmaps of translocation rates for RAD51, FANCA, BRCA2 and RAD52 depletions, each row and each column represent different loci that are either HR or NHEJ-prone with each cell being the translocation rate the two loci, delta to control siRNA. **I** Average translocation rate between different DSB loci prone to either HR or NHEJ or uncut control loci, with depletion of BRCA2, FANCA, RAD52 or RAD51, points represent each biological replicate and error bars are S.D., all statistics done relative to CTRLsi using paired, two-sided t-tests, * $p < 0.05$, ** $p < 0.01$, $n = 6$ independent biological replicates for CTRLsi and 3 for all other conditions. Source data and statistics are provided with this paper.

present in other depletions (Fig. 7D, Supplementary Fig. 7c). This could represent an alternate role for BRCA2, one such possibility is the recently described role for BRCA2 in restricting POLQ-mediated end-joining[52] or it's role in promoting RNA-dependent DNA repair[53,54], both of which are currently areas of intensive research[55–58].

## Discussion

Despite the significant role of DSB-induced mutagenesis in aging, cancer biology and other diseases, our understanding of these DSB-induced mutations is relatively limited. This is particularly prominent at the nucleotide level, with most studies using either reporter assays

or whole-genome sequencing. These approaches require high levels of damage due to their relatively low sensitivity and also suffer from additional caveats such as using exogenous loci or the lack of individual nucleotide resolution[3–6]. To address this, we developed iMUT-seq, a technique that profiles mutations at extremely high sensitivity and at single nucleotide resolution around endogenous DSBs spread across the genome, allowing for the investigation of DSB-induced mutations at a level never seen before.

At its core, iMUT-seq is a targeted sequencing approach for high depth characterisation of genomic regions. Here, we have applied this to the DIvA cell system to study mutagenesis at DSBs, however, it is also possible to apply iMUT-seq in other contexts, such as around G-quadruplex loci, replication origins, or indeed any genomic location to study mutagenesis. There are also a number of limiting factors regarding iMUT-seq, for example, common sequencing lengths only allow up to 300 bp, or ~150 bp either side of a locus. This could be remedied with further developments to sequencing technology, but will limit the applicability of iMUT-seq to smaller genomic features. Here we have multiplexed 25 loci, and although it would be possible to sequence more, there will always be a balance of breadth vs depth. Whereas 25 endogenous loci is a large increase over the single exogenous loci often used for these studies[4–6], it is still a contracted view of the genome that we are extrapolating, which may not be so broadly applicable, particularly to specific regions such as rDNA and telomeres. In our design of the primer pool, we ensured a variety of genomic loci were included, however, there is still potential for the limited number of loci to skew our results towards particular genomic features, especially as chromatin conformation and histone modifications are known to significantly impact repair[8,59]. The nature of the AsiSI induced breaks is also a limiting factor here, as the enzyme consistently induces a DSB with a 2nt 'TA' overhang, and break ends are known to influence repair mechanisms[38,60]. Furthermore, once induced, the AsiSI enzyme is able to repeatedly cut the DNA after repair prior to auxin-induced degradation, or even cut both sister chromatids simultaneously, which could have significant impacts on repair mechanics and mutation rates. Though we believe the incidence of these events is likely quite low, as the cutting efficiency of the enzyme is only ~20–30%[10,27,28], greatly reducing the probability of re-cutting the same locus or both sister chromatids being cut simultaneously. Our experimental design also has its limitations, such as the impact of the cell cycle is difficult to account for in such experiments, but is known to greatly impact DSB repair[61]. Specifically, HR is heavily regulated by the cell-cycle, being almost entirely restricted to S and G2-phase, which could influence our mutagenic signatures. Finally, although we validated certain results in different cell lines, most experiments were done in U2OS cells, which are not necessarily applicable to all cellular contexts.

We initially compared the mutations at loci prone to either NHEJ or HR alongside 53BP1 and BRCA1 depletions to elucidate repair pathway specific mutation signatures (Fig. 2). This found that NHEJ induces large deletions, likely via degradation of the exposed DNA ends, whereas HR induced base substitutions and mononucleotide deletions via polymerase error and slippage. This was an extremely interesting discovery, as HR is considered a high-fidelity pathway that incurs little or even zero mutations[6,62–64]. Interestingly, some recent reports have suggested that mutations may occur during HR, through misalignment of the resected DNA or via polymerase error, but these have mainly focused on HR-related processes, such as gene conversion or telomere maintenance, and have not yet described HR-dependent mutations at DSBs[13–15].

It should be noted that although we are using the classical models of NHEJ vs HR repair, the current literature demonstrates more complex models for repair[65]. There are many examples of crosstalk between these pathways[66–68], e.g. NHEJ can operate via a resection-dependent mechanism that is slower than the fast-kinetic, blunt ligation NHEJ[65,68,69]. This slow-kinetic NHEJ primarily uses Artemis for resection of short

tracts (0-5nt), but can also use MRE11-dependent resection tracts[68,70]. This perspective gives more clarity on some of the phenotypes we observed here, for example the shorter deletions after siRNA depletion of early resection factors (Fig. 6A, Supplementary Fig. 6a–c). This could be the result of NHEJ out-competing HR at sites already resected by MRE11. However, a more elegant possibility is that this is a shift from MRE11-dependent slow-kinetic NHEJ towards either fast-kinetic NHEJ or Artemis-dependent slow-kinetic NHEJ, as both would show reduced resection compared to MRE11-dependent slow-kinetic NHEJ.

All recent publications identifying DSB-induced mutations only focused on high frequency break-adjacent indel events, occurring at rates of over 10%, following high levels of damage, and did not study base substitutions at all or indel events that are away from the break point[5–7]. However, due to iMUT-seq's advanced sensitivity, we were able to discover extensive base substitutions and deletions that occur at distance from the break at rates as low as 0.02%, characterising an interesting mutagenic consequence of HR. Although 0.02% is a low mutation rate, this is per nucleotide of DNA, and since kilobases of DNA are resected at DSBs, these mutations become very frequent per break. Even with our limit of 100 bp either side of the break, we found distant deletions to be more common than break-adjacent deletions and distant substitutions to be the most common mutation at DSBs, making distant mutations vital in the study of DSB-induced mutagenesis. These results establish mutations introduced at distance from the DNA break as a major mutagenic consequence of DSB repair.

Some studies have suggested a potentially mutagenic aspect of HR due to the need for repolymerisation of kilobases of DNA[13–16], which is greatly supported by our results here. We find that not only does HR induce a large number of base substitutions and mononucleotide deletions up to 100 bp from the break (Fig. 6A–C, Supplementary Fig. 6a–c), but that these were directly caused by Pol-ε and to a lesser extent Pol-δ confirming this phenotype as polymerase error and slippage. It is also probable that these mutations extend beyond the 100 bp form the break that we quantify, theoretically extending up to the limit of resection. Furthermore, we determined that HR does significantly reduce translocations, despite appearing to promote all other mutation types (Figs. 3D, 6). Given the serious implications of translocations in mammalian biology[46,71], it is possible that a primary function of HR is to prevent translocations at DSBs. Interestingly, the HR-prone loci we profiled had substantially higher rates of translocations than the NHEJ-prone loci (Fig. 2I, J). Although the HR and NHEJ-prone classification of these loci is not absolute[9], this difference is still very striking. It may be that these loci are targeted for HR to counter this high translocation rate, or that an indirect connection exists between being HR-prone and translocation-prone, such as transcriptional activity[9,72,73] or physical proximity and clustering of different DSBs[10].

It is also quite remarkable that both NHEJ and HR significantly contribute to mutagenesis at DSBs, but with very different mechanisms and mutagenic signatures. Given our findings here, perhaps the key to understanding the regulation of these pathways is actually in their mutagenic signatures. NHEJ is significantly more prone to the large deletions and translocations commonly seen at DSBs[74,75], whereas HR is mostly prone to base substitutions and small deletions[13,15]. Mutations induced by HR could therefore be considered less toxic, as although small deletions can cause frameshifts in coding regions, substitutions are relatively benign. In addition, these mononucleotide deletions were via replication slippage at homopolymer stretches of DNA, and these stretches are known to skew away from coding regions or towards the ends of genes where frameshifts are less deleterious[76–80]. By comparison, the NHEJ induced large deletions and translocations can be highly genotoxic, especially in gene bodies, and HR has been previously shown to target transcriptionally active regions over NHEJ[9]. It therefore remains likely that HR preserves the integrity of the coding regions of the genome, but rather than by being an error-free process, this is by shifting mutagenesis towards less toxic

mutations and away from highly toxic mutations, particularly translocations[46–49].

These insights into repair mechanics mark a significant step forward in our understanding of DSB repair and its mutagenic consequences, and were made possible by the high sensitivity and resolution of iMUT-seq. Further study is needed to fully understand these mutagenic mechanisms; however the clear utility of this technology will be crucial for our further development towards understanding DSB-repair mechanisms.

## Methods

### Cell culture and transfection

U2OS, HCT116 and RPE-1 cells were obtained from ATCC (HTB-96, CCL-247, CRL-4000) and AID-DIvA cells were received from the Legube Lab. U2OS, AID-DIvA and HCT116 cells were all cultured in Dulbecco Modified Eagle's Medium (DMEM, GibCo) supplemented with 10% Fetal bovine serum (FBS) and 2mM L-glutamine, with AID-DIvA cell culture medium also supplemented with 800 µg/mL G418 (Formedium, G418S). RPE-1 cells were cultured in F12:DMEM (PAN-Biotech) supplemented with 10% FBS and 2mM L-glutamine. All cells were incubated at 37 °C with a 5% $CO_2$, humidified atmosphere. U2OS, AID-DIvA and RPE-1 cells are female and HCT116 cells are male. All cell lines were tested for mycoplasma contamination each time they entered cell culture from storage and were always found to be negative.

All transfections were completed using Dharmafect 1 (Horizon Discovery, T-2001-03). Dharmafect 1 was used at a final dilution of 1/1000 and siRNA (Supplementary Data 2) was used at a final concentration of 20 nM. Both Dharmafect and siRNA were separately diluted in serum free medium to a volume that was 5% of the desired final medium volume and incubated at RT for 5 mins. The siRNA and Dharmafect were then combined in a 1:1 ratio and incubated for 20 mins at RT. The siRNA dilution was then made up to the desired final volume with antibiotic free medium, and the cell culture medium was immediately replaced with this transfection medium. Cells were then incubated for 24 hours, after which the medium was refreshed with regular, antibiotic free medium. Cells were then incubated for a further 48 hours before experimental treatments began.

### Cell treatments

For DSB induction in AID-DIvA cells, treatment with 300 nM 4-hydroxytamoxifen (OHT) was given for 4 hours. For DSB repair via degradation of the AsiSI fusion protein, OHT treated cells were washed twice in pre-warmed PBS, then once in pre-warmed medium containing 500 µg/mL auxin (IAA) (Sigma, I5148) and then finally replaced with fresh medium containing 500 µg/mL auxin and incubated for 14 hours.

ATMi (KU-55933, Merck SML1109), ATRi (VE-821, Merck SML1415) and DNA-PKi (NU7441, Tocris 3712) were all used at 10 µM and administered in a 1 hour pre-treatment and maintained throughout the period of the experiment.

Etoposide treatments were completed for 1 hour with either 5 µM (IF) or 20 µM (Metaphase spreads) concentration. Cell culture medium was replaced with pre-warmed medium containing the desired concentration of etoposide, incubated for 1 hour and then cells were washed twice in pre-warmed PBS, once in pre-warmed medium and finally replaced with fresh medium and incubated for the indicated recovery period.

### Western blotting

Cells were lysed in 1.1x NuPAGE LDS sample buffer (Thermofisher, NP0007), scraped from their plates and transferred to microcentrifuge tubes. Samples were passed through a 23 gauge needle 10 times to shear DNA and homogenise the samples, and then heated to 95 C for 10 minutes. Samples were run on either 4-12% NuPage Bis-Tris gels (Thermofisher, NP0322PK2) or 10% polyacrylamide gels. Proteins transferred in tris-glycine transfer buffer containing 20% methanol and 0.05% SDS onto nitrocellulose membranes for 1.5 hours at 100 V. Membranes were then blocked in TBST containing 5% BSA at RT for 1 hour. Primary antibody probing was done overnight at 4 C with all antibodies (Supplementary Data 3) diluted in TBST containing 5% BSA. Membranes were then washed three times in TBST for 10 minutes at RT and then probed with secondary antibodies (Li-COR Biosciences) diluted 1/10000 in 5% BSA in TBST for 1 hour at RT. Membranes were then washed three times in TBST for 10 minutes at RT before scanning with a Li-COR Odyssey.

### Metaphase spreads

A total of 500,000 HCT116 or RPE-1 cells were seeded onto 10 cm plates and incubated for 24 hours before being transfected as described above. Transfected cells were incubated for 24 hours, then their medium was replaced, and they were incubated for a further 48 hours. The cells were then either treated with 2 Gy irradiation or 20 µM etoposide for 1 hour, as described earlier, and then allowed to recover for 14 hours. Metaphase cells were then enriched by treating with the microtubule poison colcemid (Sigma, D7385) at 200 nM for 1 hour.

Cells were trypsinised, pelleted, washed once in PBS and then resuspended in 10 mL of 75 mM potassium chloride. Cells were then allowed to swell by incubating at 37 °C for 30 minutes. 5 mL of ice-cold fixative (75% methanol, 25% acetic acid) was then slowly added to the cells. Cells were pelleted at 200 g, resuspended in 10 mL of fixative and incubated on ice for 2 mins twice to completely fix and wash off any residual buffer from the cells. Cells were finally pelleted and resuspended in 5 mL of fixative and dropped onto glass slides from a height of 15-20 cm using a p200 pipette. Slides were then steamed for 5 s over a water bath set to 80 °C, and then checked under a light microscope to ensure optimal spreading before drying overnight. Slides were stained with DAPI by immersing in water containing 0.1 µg/mL DAPI, washed by immersing in water and then allowed to dry overnight. Coverslips were then mounted to the slides with Vectashield anti-fade mounting medium (Vector Laboratories, H-1000). Spreads were then imaged using a Carl-Zeiss LSM 710 confocal microscope under a 63x objective and analysed in ImageJ.

### Immunofluorescence

Four thousand U2OS cells were seeded onto 12-well removable silicon chambered slides from Ibidi (Thistle Scientific, 81201) and transfected as described earlier. Cells were then treated with 5 µM etoposide for 1 hour before washing twice with pre-warmed PBS and once with pre-warmed medium and then allowed to recover for 3 hours. Cells were then pre-extracted at RT for 3 min in CSK buffer (100 mM NaCl, 300 mM sucrose, 3 mM $MgCl_2$, 10 mM PIPES pH 7.0, 50 mM NaF, 5 mM sodium orthovanadate, 10 mM β-glycerol phosphate and 0.7% Triton). Cells were washed once in CSK, once in PBS and fixed in PBS containing 2% paraformaldehyde for 10 minutes at RT. Cells were washed once in PBS, once in TBST and blocked with TBST containing 10% goat serum (Merck, G9023) for 1 hour at RT. Cells were washed twice in TBST and probed overnight at 4 °C with primary antibodies diluted in TBST containing 1% goat serum. Cells were washed 4 times in TBST for 5 mins at RT and probed with alexa-fluor conjugated secondary antibodies, diluted 1/1000 in TBST with 1% goat serum, for 1 hour at RT. Slides were washed 4 times in TBST for 5 mins at RT, dipped in water to remove residual buffer, and a coverslip was mounted using Vectashield anti-fade hard-set mounting medium containing DAPI (Vector Laboratories, H-1500). Slides were imaged using a Carl Zeiss LSM 710 confocal microscope under a 63x objective and images were analysed in Fiji[81] using the FindFoci plugin[82].

### MMEJ reporter assay

A total of 500,000 U2OS MMEJ reporter cells were seeded onto 10 cm plates and transfected with siRNA as described earlier. Forty eight hours after siRNA transfection, cells were transfected with a plasmid

encoding the I-SceI endonuclease (Addgene #31482) using Lipofecta-mine 2000 (ThermoFisher, #11668019) according to the manufacturer instructions. Cells were incubated for a further 72 hours before har-vesting by trypsinisation. Cells were pelleted at 300 $g$, washed once in PBS, re-pelleted and then analysed via flow cytometry on a Beckman Coulter CytoFLEX cytometer. Cells were first selected by FSC/SSC values, then by IFP expression of the I-SceI encoding plasmid before quantification of the GFP-positive cells to prevent variations in trans-fection efficiency from impacting our results (Supplementary Fig. 7e).

### iMUT-seq experimental protocol

All iMUT-seq experiments were done in three biological replicates that were independently carried out. 60,000 AID-DIvA cells were seeded into 6-well plates, incubated for 24 hours and transfected as described earlier with siRNA from Supplementary Data 2. Transfected cells were incubated for 24 hours, their medium was replaced, and they were incubated for a further 48 hours. Cells were treated with or without 300 nM 4-hydroxytamoxifen (OHT) for 4 hours to induce DSBs, washed twice with pre-warmed PBS and once with pre-warmed med-ium containing 500 μg/mL IAA, and replaced with media containing 500 μg/mL IAA and incubated for 14 hours to degrade the AsiSI fusion protein and allow complete DSB repair. Cells were placed on ice, washed once in PBS and lysed in cytoplasmic lysis buffer (50 mM HEPES pH7.9, 10 mM KCl2, 1.5 mM MgCl2, 0.34 M sucrose, 0.5% triton, 10% glycerol, 1 mM DTT) for 10 minutes. Cells were washed once in cytoplasmic lysis buffer and the nuclei were lysed in genomic extrac-tion buffer (50 mM Tris pH 8.0, 5 mM EDTA, 1% SDS, 0.5 mg/mL Pro-teinase K). The nuclear lysates were transferred to 2 mL DNA LoBind microcentrifuge tubes (Fisher Scientific, 0030108426) and incubated in a thermomixer at 60 °C for 40 minutes with 500 rpm agitation. 0.1 volumes of 3 M sodium acetate pH 5.2 was added followed by 2.5 volumes of 100% ethanol, the tubes were inverted several times to mix an then incubated on ice for 1 hour to precipitate the genomic DNA. The DNA was pelleted at 19000 g for 20 minutes at 4 °C, washed in 75% ethanol and re-pelleted for 10 minutes twice. The ethanol was aspi-rated and the DNA pellet allowed to dry at RT before resuspended in water and quantifying the DNA concentration via nanodrop.

For each condition, 3 50 μL PCR reactions were run each con-taining 0.7 μL Phusion polymerase (NEB, M0530L), 1X Phusion HF buffer, 2 M betaine, 1.5% DMSO, 400 μM dNTPs, multiplexed 25 genomic primer pairs (Supplementary Data 1) at a concentration of 40 nM per primer, i.e. 2 μM total, 1 μg genomic DNA. The reaction was carried out as follows: 98 °C 5 mins, 12 cycles of 98 °C 60 s, 62 °C 120 s, 72 °C 120 s, then 72 °C 5 mins. All 3 reactions were combined and 100 μL of the total reaction volume was taken forward for size selec-tion. 60 μL of SPRISelect beads (Beckman, B23318) was added to the 100 μL PCR reaction mixture, mixed by pipetting and then incubated for 5 mins at RT. The beads were collected on a magnetic rack and the supernatant was transferred to a fresh tube. 70 μL of beads were added to the supernatant and mixed by pipetting and incubated for 5 mins at RT, binding the amplicons. Beads were collected on the magnet, the supernatant was discarded and the beads were washed twice in 85% ethanol for 30 s. The beads were dried until most ethanol had evapo-rated, but the beads were still wet, and the beads were then resus-pended in 50 μL 10 mM Tris pH 8.0 and incubated for 10 minutes at 37 C with 1000 rpm agitation. The 50 μL of eluted amplicons were then transferred to a fresh tube and the process of bead selection was repeated but with half the volumes (30 μL and 35 μL of beads) and eluted in 25 μL of 10 mM Tris pH 8.0. 10 μL of these amplicons were used for final library preparation using the NEB Ultra II DNA library prep kit according to the manufacturer's instructions using a 1/25 adapter dilution and 5 PCR cycles.

These final libraries were then quantified using the Qubit dsDNA HS kit (Thermo, Q32854), pooled and sequenced on a NextSeq 500 using a high output 300 cycle kit set to run paired-end 150 cycles.

### Traffic Light Reporter Sequencing

For mutation profiling of the TLR reporter, 500,000 cells were seeded on 10 cm plates for each condition. siRNA transfection was carried out as described earlier and cells were incubated for 48 hours. Transfec-tion of a plasmid encoding the I-SceI endonuclease (Addgene #31482) as well as the donor plasmid (Addgene #31485) was carried out using Lipofectamine 2000 (ThermoFisher, #11668019) according to the manufacturer instructions. For uncut samples, only the donor plasmid was transfected. Cells were incubated for a further 72 hours, after which DNA extraction, library preparation and sequencing was carried out as described for regular iMUT-seq with one addition. Initial PCR amplification was done for the 5 uncut iMUT-seq control amplicons as well as a 1155 bp amplicon around the I-SceI cut site in the TLR reporter cassette. This was done to target the genomic TLR locus specifically, without amplifying the exogenous template that had been transfected into the cells. This amplicon was then digested with the restriction enzymes Afl-II, Bts-I and Alw-I (NEB, #R0520, #R0667 and #R0513 respectively) for 4 hours at 37 C with 300 rpm agitation in 1x CutSmart buffer (NEB). Following this, amplicons were size selected and library preparation was completed as described for standard iMUT-seq. Fastq files for each sample were divided into HR-positive and HR-negative files using a python script that identified the corrected eGFP gene using the sequence "GAGGGCGAGGGCGATGC" or it's reverse com-plement. Analytical mutation profiling was also carried out as for standard iMUT-seq.

### iMUT-seq translocation mapping

Translocation mapping was conducted on raw fastq files using our custom tool mProfile TransloCapture (Available at https://github.com/aldob/mProfile and for install via https://pypi.org/project/mProfile-mut/). TransloCapture uses the sequences of the genomic primers initially used to amplify our target sequences to identify which primers were used to amplify each sequencing read, deter-mining if the read is from an accurately repaired or a translocated DSB and which sites were translocated together. TransloCapture also allows the non-translocated and the translocated reads to be separate from the other reads and both to be written separately to new fastq files. TransloCapture outputs translocation map tables which were then used for all downstream translocation analysis via custom python and R scripts (Scripts available at https://github.com/aldob/iMUT-seq[83]).

### iMUT-seq mutation profiling

Exact parameters and settings used for raw data processing can be found in the raw data pipeline shell script (Scripts available at https://github.com/aldob/iMUT-seq[83]). First, mProfile TransloCapture was used to filter out any translocated reads from the fastq files and Fastp[84] was used to filter out low-quality reads prior to alignment with Bowtie 2[85]. We noticed that DSB-induced samples had significantly reduced alignment efficiency due to their mutations. To address this, we wrote a custom machine learning process using a genetic algorithm (Scripts available at https://github.com/aldob/iMUT-seq[83]) to systematically test different Bowtie2 alignment parameters, optimising the efficiency (Supplementary Fig. 1b). This yielded the following parameters: "--fr --maxins 400 --no-discordant --no-mixed --ignore-quals --no-1mm-upfront -D 100 -R 50 -L 28 -N 1 --np 0 --dpad 49 --gbar 2 --mp 3.2,0.35 --rdg 1,1 --rfg 5,2 --score-min L,−1.0,−0.5" which significantly increased alignment efficiency and removed the difference between undamaged and damaged samples (Supplementary Fig. 1c). Alignments were then sorted using Samtools[86] sort and indexed using Samtools index. Raw mutation calls were then generated using Samtools mpileup. These raw mutation calls were then parsed by our custom tool mProfile callMUT (Available at https://github.com/aldob/mProfile and for install via https://pypi.org/project/mProfile-mut/) into mprofile mutation call files.

Mprofiles were then used for all downstream mutation analysis and quantification via custom python and R scripts (Available at https://github.com/aldob/iMUT-seq[83]).

## iMUT-seq mutation quantification

All mutations are calculated as a percentage of reads at the nucleotide position that the mutation occurs. A delta of damaged-undamaged mutation rates was used to remove background mutations that are either naturally present within the genomes of our cells or that were induced via PCR or sequencing error. This delta was conducted on a per-nucleotide level, subtracting the rate of each mutation type at each individual genomic position in the undamaged sample from the rate of that mutation type at the corresponding position in the damaged sample. This generated a DSB-induced mutation profile for each condition. Where results are shown relative to control siRNA treatment, this same approach was taken to subtract the mutation rates for the damaged-undamaged delta of the control siRNA from the damaged-undamaged delta of the treatment siRNA at each nucleotide sequenced.

The mutation profiles, in the form of mprofile files, were then used either for directly generating metagene line plots of the mutation profiles, or to create average overall mutation rates. Where average mutation rates are used, this is calculated per site i.e. the average total mutation rate across the regions sequenced per DSB quantified (R and python scrips for this are available at https://github.com/aldob/iMUT-seq[83]).

## Statistics & reproducibility

All experiments were done in biological triplicate, with each replicate being cultured and treated on separate days using a different cell passage. iMUT-seq experiments were done in 2 phases to split up the conditions, therefore CTRLsi results have 6 biological replicates as each replicate across both phases of experiments had matched CTRLsi samples. No statistical method was used to determine sample sizes, 3 biological replicates per experiment was chosen as it is the common convention for biological studies for finding statistically relevant results. No data was excluded from these studies. No randomisation was used in the experimental design. For metaphase spread and immunofluorescence experiments, samples were blinded after processing but before data collection and the blinds were only released after data analysis was complete.

For statistical analysis, where average values per replicate were being compared, paired t-tests between the replicate values for each condition were used. For testing distributions of multiple data points, for example mutation rates between different groups of loci or fluorescent foci per nucleus between different conditions, unpaired Wilcoxon tests were used to provide a non-parametric comparison of the data distributions.

## Reporting summary

Further information on research design is available in the Nature Portfolio Reporting Summary linked to this article.

# Data availability

All iMUT-seq raw data has been deposited at ArrayExpress under the accession "E-MTAB-11259". Source data, including statistics, are provided with this paper. Source data are provided with this paper.

# Code availability

All analytical code is publicly available on GitHub (https://github.com/aldob/iMUT-seq[83]), which includes the raw data processing pipeline and it's parameters as well as the code used to generate plots in R. All relevant software tools/packages are listed in the relevant methods sections along with the versions used.

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

## Acknowledgements

We thank Cancer Research UK for their core funding to the CRUK Beatson Institute A17196 and A31287 and for core funding for the Bushell lab (A29252). This work was also supported by Cancer Research UK RadNet Cambridge [C17918/A28870]. We also thank Gaëlle Legube and their lab for gifting us the AID-DIvA cells and for their continual support throughout the project. Finally we would like to thank the Steve Jackson Lab for gifting the U2OS-TLR and U2OS-MMEJ reporter cell lines and for the resources for those experiments.

## Author contributions

A.S.B. developed iMUT-seq, conducted all experiments and bioinformatic analysis and wrote the manuscript. A.S.B. and M.B. both conceived the technique and contributed to manuscript drafting.

## Competing interests

The authors declare no competing interests.
