## [Peer Review File · Nature Communications]

REVIEWER COMMENTS

Reviewer #1 (Remarks to the Author):

In their manuscript, Bader and Bushnell describe a method to measure scars at AsiSI-induced DNA breaks using a multiplex PCR method and deep sequencing. This enzymes cuts at endogenous loci, allowing to study repair scars after DNA break induction. The major new insight is that repair using NHEJ and HR is mutagenic, and the main function of HR appears to block translocations.

Overall, the take-home message is provocative, and the method is effective and adds to the growing list of NGS-based approaches to measure DNA repair. A shortcoming of the manuscript is that usage of HR or NHEJ is based on a previous study in which some loci were shown the preferentially be repair by either NHEJ/HR, which is not strict. Furthermore, some of the conclusions contain hypothetical models, which are not extensively tested (eg mutations upon HR usage is due to error-prone fill in at resected DNA and roles for FANCA in rad51 and BRCA2 in RPA recruitment, based on a single time point; this could also reflect different repair dynamics).

Specific comments:

-It would be good to confirm the mutational scars upon HR usage using an independent source of DNA breaks.

-Figure 1D: the lower left panel is called 'accurate repair', but the result indicated non-accurate repair, albeit that the proper DNA ends is used. Using a 'cis' versus 'trans' DNA end for repair is a better description than 'accurate repair' vs 'translocated'

- A shortcoming of this assay (and other nuclease or cas9-based methods) is that both sister chromatids at a specific locus may get cut in S-G2 cells, which may lead to different lesions that spontaneous breaks affecting only one sister chromatid. Potentially, this may influence or prevent template-based HR and may underlie translocations that are attributed to HR (Fig 2H). perhaps the subsets of HR-prone loci, are the sites where AsiSI cutting is most efficient, targeting both sisters.

- AsiSI creates a large number of DNA breaks, likely also leading to cell cycle arrest. How is cell cycle status (which affects repair choice) taken into consideration in HR of NHEJ usage? -

- HR-prone loci show increased numbers of translocations upon AsiSI-induced DSBs. How do the authors explain this apparent discrepancy with observations that depletion of HR factors promote translocations (fig. 6G) and the observations in the field that HR mutations lead to increased frequencies of translocation?

-The description of 'loss of chromosomes' and 'large-scale genomic destabilization' and 'chromosome scale deletions' are rather vague. In case of DSBs, one would expect that parts of chromosomes can get missegregated during cell division, leaving centromere-containing parts. Can the authors provide a more detailed explanation?

- Overall, the text could be improved by use of commas, and hyphens. Line 186: doesz ->does
- Concerning the quantification of mutation rates: the analysis is done from bulk cultures, involving a multiplex PCR step. The data in my opinion provide a list of possible outcomes, rather than a quantitative outcome. How can one distinguish between non-cut and error-free repair?
- AsiSI creates TA overhang. Good to mention that the repair is measured in this specific context.

Reviewer #2 (Remarks to the Author):

Bader and Bushell iMUT-seq: high resolution DSB induced mutation profiling reveals prevalent homologous recombination dependent mutagenesis.

In this manuscript, the authors describe iMUT-seq, a technique that profiles DSB-induced mutations at high sensitivity and single-nucleotide resolution around endogenous DSBs. By doing so in various repair mutant backgrounds the authors find that homologous recombination is highly mutagenic, resulting in both prevalent base substitutions as well as deletions while translocations are reduced. The authors suggest that these errors arise from DNA polymerase errors.

The manuscript is well written, the data are clear and well presented, and on the face of it the manuscript applies what it claims to be a novel technology (iMUT-seq) to the analysis of DSB repair profiles in particular mutant backgrounds. In doing so they claim to have made a number of revealing and remarkable insights.

General comments:

1. The manuscript begins by describing the advantages of iMUT-seq. iMUT-seq does provide the advantage of being able to analyse the repair and misrepair of multiple DSBs simultaneously around a 100bp sequence surrounding the DSB, which can capture translocations. Further, they address the enzymatic recutting problem by employing a rapidly degradable AsiSI enzyme. All good. However, there are some disadvantages to the approach too. One problem with iMUT-seq over other technologies, not highlighted in this manuscript, is that it requires prior knowledge of where the break site is for PCR amplification across these sites. In using the AsiSI approach much of their work is necessarily limited to the analysis of a single cell line, which is problematic for two reasons. First the AsiSI enzyme has been reported to cut up to 150 times in a single cell, which has the potential to compromise the repair of an individual DSB (not tested). Moreover, by limiting the analysis to one cell line it is difficult to infer the generality of the findings.

2. While the figures do, the manuscript does not include the rate of mutations observed in the various mutations compared to controls (where possible), leading the reader to continually refer to the figures. Including these figures in the text would be helpful.

3. Related to the above point and without the exact figures being presented in the text, the authors suggest that HR is 'highly mutagenic'. However, at its highest, mutations (substitutions or deletions) do not rise above 0.03%. that means of course that HR is accurate in 99.97% of reads (per nucleotide). Does this not in fact suggest that HR is in fact highly accurate?

4. Conversely the data demonstrates that NHEJ- prone repair is similarly low at ~ 0.01% of reads per nucleotide. This contrasts with the prevailing view that in fact NHEJ is error prone. Yet the authors do not make much of this point in the manuscript, which they potentially could.

5. The role of HR in preventing translocations is quite important and to my knowledge novel. It would be important to know what the underlying mechanism leading to the translocations is.

6. Similarly, the observation that disrupting XRCC4 leads to increased large deletions is of interest and again the mechanisms underlying this process have not been determined.

7. P6 and p11: Here the authors investigate the impact of knocking down 53BP1 and BRCA1 respectively. 53BP1 depletion was found to promote distant deletions while BRCA1 depletion had the opposite effect. The authors claim that this suggests that break-distant mutations were observed are mostly HR-dependent which may suggest is due to polymerase error and slippage during repolymerisation. The authors have followed through on this and have knocked down POLD or POLE, and have found that this reduces the number of mutations, leading to the conclusion that these polymerases must be error prone. While this may be the case, the authors fail to exclude the possibility that these mutations have been introduced as a result of accurately copying mutated templates such as the homologous chromosome or pseudogenes? Knockdown of POLE would be expected to reduce this mechanism of mutagenesis too. as is observed. This possibility needs to be excluded before it can be concluded that POLE is mutagenic.

8. The analysis of BRCA2 and FANCA to RAD51 loading, while of interest, is not really to do with the mutations associated with DSB repair and feels like it should be part of a separate study.

9. In contrast, the role of BRCA2 in resection, which is perhaps more related to the manuscript focus is not followed up.

Specific points:

P9: line 186: doesz to does.

P12: 264: knockdown of FANCA

P12: 266: similar to the RAD51 profile.

Reviewer #3 (Remarks to the Author):

Summary:

In this study the authors introduce a new technique for analysing the mutations arising at DSBs generated in the Diva system developed by Gaelle Legube, a system involving the induction of DSBs at a subset of AsiS1 sites in the genome of U2OS cells (which should be discussed in the introduction with respect to its strengths and weaknesses). Bader et al perform NGS at such depth that they can catalogue mutations that occur during repair of this subset of sites. Importantly, their study exposes the mutagenic nature of repair by homologous recombination. HR is often considered “error-free” and the work of Bader and Bushell challenges this assumption, at least for HR-dependent repair at a subset of AsiS1 sites. They should point out that HR-dependent repair at other DSBs (e.g. DSBs generated during S phase) might not be so mutagenic. In fact, the persistent ‘cut and repair’ cycle likely to occur at each cut site until the enzyme is removed may select for mutations. Thus, the authors should not overinterpret their interesting results with the DivA. In addition, their study supports previous data (which should be cited) that HR might function primarily to reduce translocations rather than mutations.

Major Points:

1. The introduction is admirably succinct and discusses the shortcomings of many technological approaches to DSB repair. However, although DiVA system developed by the Legube laboratory many

years ago has been very useful to the field, they should also present the clear shortcomings of this system in their introduction.

2. The chromatin environment is a critical consideration with respect to the AsiSI-induced DSBs in the DiVA system. They are believed to occur in euchromatic/transcribed regions, while AsiSI sites that remain uncut are predominantly located in heterochromatic or poorly transcribed regions. The authors should include much more discussion, as well as interpretation, of the chromatin environment of the breaks, as well as speculation about how their results may extrapolate to DSBs at other genomic locations (e.g. heterochromatin, centromeric, telomeres, repetitive regions, fragile sites, rDNA, etc).

3. Figure 3A. The “NHEJ” and “HR” categorisation needs to be revised because MRE11 and ATM/ATR regulate both these pathways. Remove these factors from the “HR progression” wedge and include in their own category at the beginning of the plot. Also, 53BP1 function upstream of Artemis, and RAD52 function in SSA not gene conversion and if included should be placed at the end of the plot (recently reviewed in Kieffer and Lowndes 2022).

4. Also, please comment on why the authors have used chemical inhibition instead of siRNA depletion for some of the proteins (e.g. in Figure 3A)? It would have been better to use siRNA depletion for all, this is especially important as pharmacological inhibition of PIK kinases may ‘lock’ these kinases onto DSBs thereby preventing accurate DSB repair. This is a quite different scenario than depleting PIK kinases and no doubt explains their result with DNA-PKi which, confusingly, in highly elevate mutation rate. It would be better to remove the PIKK inhibitor data from the manuscript and, if not, replace it with depletion data.

5. The use HCT116 cells in Figure 4F-H and 5H-I) is a bad choice as these MIN cells are known to have elevated levels of HR. The authors should repeat this analysis with more normal cells, e.g. RPE1 cells might be suitable. In this case, the authors should also consider using ICRF193 instead etoposide because the former is believed to result in a higher proportion of breaks being repaired by NHEJ.

6. Figure 6. It is inappropriate to use MRE11 depletion and ATM/Ri as readout for HR (see above). Retitle this figure and amend text accordingly.

7. Figure 7. It is not surprising that RAD52 differs from BRCA2 and FANCA as it is primarily an SSA factor (see above). Retitle this figure and amend text accordingly.

8. Include in Discussion text addressing why the analyses were limited to just 100bp either side of the AsiSI sites as this ‘distance’ is still proximal given that resection can occur over much greater ‘distances’.

9. Include in Discussion text on the possibility of being able to perform similar analyses on other known genomic sites (e.g. G4s, sites of transcription/replication clashes, very highly expressed genes) upon further improvements to the required technology.

Conclusion:

All in all, this study is well structured, thought-through adds provides new insight into DSB repair in human cells. However, the authors need to be careful to avoid over-interpretation. Furthermore, they should discuss in detail the limitations of the system used, the nature of the chromatin environment at the breaks and speculate on the likely mutagenic outcome at 'distance' further than 100bp away from the break (up to 3.5 Kb, Zhou et al (2013) NAR, Paull lab).

Minor points:

1. Supplemental Figure 2 legend is very confusing. If H is the same as E which is the same as C, would then H not be the same as C? Please clarify. Also state reasoning for using different types of statistics in different parts of Figure. Similarly for other Figure legends.

2. Line 143/144. The authors claim that NHEJ induces large deletions around the break, where are the large deletions demonstrated and please define what is meant by 'large' in this case?

3. Use the word 'depletion' instead of 'knockdown' throughout as 'knockdown' is technical jargon.

4. Figure 4A in graph legend (coloured squares) is missing

5. Line 186 "does" is spelled incorrectly

6. For microscopy images throughout grey scale images give better contrast and may well better present the data.

7. Clearly state sequencing depths in main text.

Reviewer #1 (Remarks to the Author):

In their manuscript, Bader and Bushnell describe a method to measure scars at AsiSI-induced DNA breaks using a multiplex PCR method and deep sequencing. This enzymes cuts at endogenous loci, allowing to study repair scars after DNA break induction. The major new insight is that repair using NHEJ and HR is mutagenic, and the main function of HR appears to block translocations. Overall, the take-home message is provocative, and the method is effective and adds to the growing list of NGS-based approaches to measure DNA repair. A shortcoming of the manuscript is that usage of HR or NHEJ is based on a previous study in which some loci were shown the preferentially be repair by either NHEJ/HR, which is not strict. Furthermore, some of the conclusions contain hypothetical models, which are not extensively tested (eg mutations upon HR usage is due to error-prone fill in at resected DNA and roles for FANCA in rad51 and BRCA2 in RPA recruitment, based on a single time point; this could also reflect different repair dynamics).

Specific comments:

-It would be good to confirm the mutational scars upon HR usage using an independent source of DNA breaks.

We agree an orthogonal validation for the HR mutational signature would be very beneficial to the manuscript. To address this, we used the traffic light reporter (TLR) assay for homologous recombination which uses a transfected exogenous template to correct a mutated GFP gene after DSB induction via the I-SceI endonuclease (Supplemental Fig. 6d). By sequencing over the break, we could segregate reads into groups that were either sites that had been repaired by HR or not, due to the sequence corrected by HR.

This approach identified a similar mutation signature to our previous results (page 12, paragraph 3, Supplemental Fig. 6f-h), and we believe this approach, although using a reporter assay, is more specific to HR as alternative resection dependent repair mechanisms, such as MMEJ and SSA, would not introduce the HR corrections here.

Thank you for the suggestion, we think this has significantly improved the manuscript.

-Figure 1D: the lower left panel is called 'accurate repair', but the result indicated non-accurate repair, albeit that the proper DNA ends is used. Using a 'cis' versus 'trans' DNA end for repair is a better description than 'accurate repair' vs 'translocated'

We agree this is confusing, we have adjusted this figure as requested.

- A shortcoming of this assay (and other nuclease or cas9-based methods) is that both sister chromatids at a specific locus may get cut in S-G2 cells, which may lead to different lesions that spontaneous breaks affecting only one sister chromatid. Potentially, this may influence or prevent template-based HR and may underlie translocations that are attributed to HR (Fig 2H). perhaps the subsets of HR-prone loci, are the sites where AsiSI cutting is most efficient, targeting both sisters.

We have calculated cutting efficiency of the loci we amplified from the BLESS dataset published by the Legube Lab and compared it to our mutation frequencies (figure below). We find no correlation between mutation rate and relative cutting efficiency. It is worth noting that although it is possible that both sister chromatids are simultaneously cut, since cutting frequency of each loci is ~20-30%, the chance of both chromatids being cut

simultaneously is quite low. This is an important caveat however, and we have added this to the manuscript when discussing the limitations of iMUT-seq (Page 15, paragraph 2).

- AsiSI creates a large number of DNA breaks, likely also leading to cell cycle arrest. How is cell cycle status (which affects repair choice) taken into consideration in HR of NHEJ usage?

We have previously attempted multiple cell cycle resolved experiments to explore this aspect of repair. However, we have been unable to hold the cells in a specific cell cycle phase for the duration of these experiments without impacting on the repair processes themselves, e.g. via cell-cycle inhibitors or thymidine treatment. In addition, we do not believe a synchronisation of the cells will work due to the duration of the experiment, for example, synchronising the cells and treating them in G1 phase with OHT for 4 hours followed by IAA for 14 hours would result in the cells progressing through S-phase during the experiment. Especially since U2OS cells have ineffective cell cycle checkpoints.

If the reviewer has a suggested methodology that could help to study this feature we would be happy to attempt such experiments. For now we have included a discussion of the role the cell cycle may play in mutagenesis and how this will impact on the applicability of our findings in different biological contexts (page 15 paragraph 2).

- HR-prone loci show increased numbers of translocations upon AsiSI-induced DSBs. How do the authors explain this apparent discrepancy with observations that depletion of HR factors promote translocations (fig. 6G) and the observations in the field that HR mutations lead to increased frequencies of translocation?

We do not believe that HR repair is actually resulting in these translocations, our observation is that HR prone sites are also translocation prone, but it is unclear if there a causal link between the mechanisms of HR and the mechanisms of translocation. We show that high translocation loci also show high DSB clustering (Fig. 2k), which has been previously described and was hypothesised to drive translocations⁽¹⁾, so this is a potential mechanism for the high translocation rate at these loci, however this process is currently not well understood and is actively under investigation⁽²⁾.

The HR/NHEJ-prone classification of loci is not black and white for many reasons, therefore our interpretation of HR-mutagenesis is primarily via the use of HR-factor depletions in Fig. 6. We actually describe translocations as the key type of mutation that HR protects against (Fig. 3d, Fig. 6g). It is possible that our observation of loci prone to translocation also being

HR prone (Fig. 2i) is actually a protective mechanism, though this is just speculation which is why we have not yet included this in the manuscript. We have expanded our discussion to include more depth on this phenomenon (Page 18, paragraph 1).

-The description of 'loss of chromosomes' and 'large-scale genomic destabilization' and 'chromosome scale deletions' are rather vague. In case of DSBs, one would expect that parts of chromosomes can get missegregated during cell division, leaving centromere-containing parts. Can the authors provide a more detailed explanation?

Thank you for this comment, we agree our use of terminology here is not very clear. We have adjusted the text to only use the phrase "loss of genome integrity" and then immediately described this in specific terms (page 9 paragraph 4).

-Overall, the text could be improved by use of commas, and hyphens. Line 186: doesz ->does

We have attempted to address this throughout the text.

-Concerning the quantification of mutation rates: the analysis is done from bulk cultures, involving a multiplex PCR step. The data in my opinion provide a list of possible outcomes, rather than a quantitative outcome. How can one distinguish between non-cut and error-free repair?

We agree that an absolute quantification of mutation rates will not be accurate due to a variety of factors, mainly the cutting efficiency which we have noted and expanded upon (page 7 paragraph 2).

-AsiSI creates TA overhang. Good to mention that the repair is measured in this specific context.

We have added this factor into our description of the technique (page 4 paragraph 3) and into our discussion (page 15 paragraph 2).

Reviewer #2 (Remarks to the Author):

Bader and Bushell IMUT-seq: high resolution DSB induced mutation profiling reveals prevalent homologous recombination dependent mutagenesis.

In this manuscript, the authors describe iMUT-seq, a technique that profiles DSB-induced mutations at high sensitivity and single-nucleotide resolution around endogenous DSBs. By doing so in various repair mutant backgrounds the authors find that homologous recombination is highly mutagenic, resulting in both prevalent base substitutions as well as deletions while translocations are reduced. The authors suggest that these errors arise from DNA polymerase errors.

The manuscript is well written, the data are clear and well presented, and on the face of it the manuscript applies what it claims to be a novel technology (iMUT-seq) to the analysis of DSB repair profiles in particular mutant backgrounds. In doing so they claim to have made a number of revealing and remarkable insights.

General comments:

1. The manuscript begins by describing the advantages of iMUT-seq. iMUT-seq does provide the advantage of being able to analyse the repair and misrepair of multiple DSBs simultaneously around a 100bp sequence surrounding the DSB, which can capture translocations. Further, they address the enzymatic recutting problem by employing a rapidly degradable AsiSI enzyme. All good. However, there are some disadvantages to the approach too. One problem with iMUT-seq over other technologies, not highlighted in this manuscript, is that it requires prior knowledge of where the break site is for PCR amplification across these sites. In using the AsiSI approach much of their work is necessarily limited to the analysis of a single cell line, which is problematic for two reasons. First the AsiSI enzyme has been reported to cut up to 150 times in a single cell, which has the potential to compromise the repair of an individual DSB (not tested). Moreover, by limiting the analysis to one cell line it is difficult to infer the generality of the findings.

We agree this is a significant limitation of our work here and we have added a clear section to the discussion (page 15 paragraph 2) to highlight the limitations of our approach and the impact they may have on our findings and their broader applicability.

2. While the figures do, the manuscript does not include the rate of mutations observed in the various mutations compared to controls (where possible), leading the reader to continually refer to the figures. Including these figures in the text would be helpful.

Thank you for this comment, we agree that it would be significantly more clear and informative to add data values to the text. We initially added mutation rate percentages as they are reported in the figures, however we found this to be quite unhelpful and needed additional context when discussing the effects of siRNA depletions. Instead, we have therefore opted to use fold change relative to control siRNA when referring to the results of depletions, except for Fig. 2 where we are looking at the results without any siRNA depletions and we therefore used mutation rate percentages. We have added this in throughout the results section.

3. Related to the above point and without the exact figures being presented in the text, the authors suggest that HR is 'highly mutagenic'. However, at its highest, mutations (substitutions or deletions) do not rise above 0.03%. that means of course that HR is accurate in 99.97% of reads (per nucleotide). Does this not in fact suggest that HR is in fact highly accurate?

The mutation rates of 0.03% are per nucleotide, but the mutation rate per DSB is significantly higher. Our estimate for mutations per DSB is ~1.0-1.5% (page 7 paragraph 3) which we agree still suggests DSB repair is in general quite efficient and not as mutagenic as one might expect.

As for the description of HR as highly mutagenic, we have made this claim based on the relative frequencies of HR induced mutations to the total number of mutations we observe. Depletion of MRE11 reduces distant small deletions by ~75% and these deletions are the most common deletion at DSBs, significantly more frequent than all break-adjacent deletions combined which have always been the focus of mutagenic studies into DSBs. Relative to the total mutations we observe at these breaks, HR appears to contribute a

significant level of these mutations, and based on our data this is possibly even as high as NHEJ mutagenesis which is consistently referred to as an error-prone repair pathway. Below is a figure of the total mutations at HR and NHEJ prone loci after KU70 or MRE11 depletions. In total, the average mutation rate per DSB was reduced by 46% with KU70si and by 42% with MRE11si; however, it is clear from the figure there are differences between loci, we also show in the manuscript that these depletions display different mutation signatures. We also know NHEJ is generally more active than HR, and in Fig. 4 we see that KU70 depletion doesn't give completely accurate results. As a result it is difficult to compare the absolute mutation rates of the repair pathways. The next point also relates to this regarding the observed rate of NHEJ induced mutations, and we have added a discussion on the mutagenic rates of both HR and NHEJ to the discussion (page 18 paragraph 2).

4. Conversely the data demonstrates that NHEJ- prone repair is similarly low at $\sim 0.01\%$ of reads per nucleotide. This contrasts with the prevailing view that in fact NHEJ is error prone. Yet the authors do not make much of this point in the manuscript, which they potentially could.

Overall we have found it difficult to define the absolute mutagenic contribution of each repair pathway, as complete depletion of the initiation of one pathway forces repair through the other pathway, shifting the mutational signature. As a result we have tried to keep our terminology to relative terms when discussing the mutagenesis of HR and NHEJ. One observation appears to be that NHEJ is not as mutagenic as previously thought and HR appears to be more mutagenic, especially when comparing the two pathways. NHEJ is considered error-prone and HR is often referred to as "error-free", but if anything their absolute levels of mutagenesis, as far as we can determine from our data, are actually quite similar. We believe the key difference between the mutagenesis of these pathways, and

what should be the focus, is actually in the type of mutations they are prone to. Whereby HR promotes base substitutions and mononucleotide deletions, while NHEJ promotes large deletions and translocations which are theoretically more genotoxic.

As stated in the response to comment 3, we have added a comparison of the mutagenicity of NHEJ and HR to the discussion (page 18 paragraph 2). We think this is actually a very beneficial comparison that we had not fully delved into previously, so thank you for raising these comments.

5. The role of HR in preventing translocations is quite important and to my knowledge novel. It would be important to know what the underlying mechanism leading to the translocations is.

Though our experiments here demonstrate a specific role for HR in guarding against translocations, we however believe the concept of HR deficiency leading to increased translocations is not particularly novel, for example with BRCA1 or ATM mutants/knockouts. We have now cited and discussed this data (page 13 paragraph 2, page 18 paragraph 1) which we believe has added further mechanistic understanding of our data and improves clarity for the reader.

6. Similarly, the observation that disrupting XRCC4 leads to increased large deletions is of interest and again the mechanisms underlying this process have not been determined.

We hypothesised in our manuscript that since these large deletions primarily showed microhomologies that this was due to a role of XRCC4 in preventing MMEJ and we agree this interesting observation should be investigated further. To test our hypothesis we initially conducted immunofluorescence to quantify the recruitment of GFP tagged Polymerase- θ (POLQ), a key regulator of MMEJ. We had hoped to measure POLQ-GFP recruitment under different conditions as a readout of MMEJ, however we found that in our hands POLQ-GFP signal was not quantifiable even under conditions that are known to induce high levels of MMEJ. We also extensively tested various sample preparation approaches for POLQ-GFP and tested both traditional IR-IF and the clustered-DSB inducible FokI system.

As a result, we followed up using an MMEJ reporter assay that uses an I-SceI inducible DSB flanked by microhomologies that if repaired via MMEJ results in an active eGFP gene that can be quantified via flow cytometry. We have used this assay with combinations of MRE11, XRCC4 and LIG4 depletions to shed more light on the mutagenic signatures we observed in combination with current literature on the subject (Supplemental Fig. 5g-h, page 10 paragraph 3/page 11 paragraph 1). These results show that XRCC4 depletion drives MMEJ in an MRE11-dependent manner, indicating a role for XRCC4 in anti-resection or gap-filling processes, whereas LIG4 depletion had a limited increase in MMEJ that was not entirely MRE11-dependent.

7. P6 and p11: Here the authors investigate the impact of knocking down 53BP1 and BRCA1 respectively. 53BP1 depletion was found to promote distant deletions while BRCA1 depletion had the opposite effect. The authors claim that this suggests that break-distant mutations were observed are mostly HR-dependent which may suggest is due to polymerase error and slippage during repolymerisation. The authors have followed through on this and have knocked down POLQ or POLE,

and have found that this reduces the number of mutations, leading to the conclusion that these polymerases must be error prone. While this may be the case, the authors fail to exclude the possibility that these mutations have been introduced as a result of accurately copying mutated templates such as the homologous chromosome or pseudogenes? Knockdown of POLE would be expected to reduce this mechanism of mutagenesis too. as is observed. This possibility needs to be excluded before it can be concluded that POLE is mutagenic.

This is an interesting concept, it is possible that mutations are identified as a result of alternative alleles on the homologous chromosome, which would be loss of heterozygosity (LOH). LOH is a known outcome of HR repair, but would not be polymerase error and would also be less likely to be toxic, as this allele is already present in the cell. To investigate this, we identified all sites within our sequencing that had heterozygous alleles i.e. the sequencing of the untreated cells showed two different nucleotides. However, this only yielded two loci and both of these show relatively low base substitution rates and even a shift away from the alternative allele (see figure below). Therefore we do not believe this mechanism is the likely source of our observed signature.

The possibility of alternative templates such as pseudogenes is also quite interesting, it is possible that the homology search during HR identifies a locus that is homologous but not an identical copy like the sister chromatid, resulting in the incorporation of mutations. Pseudogenes are a good example of this, as they are genomic regions that are close copies of other genomic regions, however it should be noted that in our design of these experiments we selected 20 ASIS loci that vary greatly in their genomic features, including location within genes. Of the 20 loci, 5 are not within genes, 2 of which are also prone to HR. These loci also show an induction of distant substitutions that are MRE11 dependent (see figure below). This suggests that pseudogenes are not the source of these mutations either. It is very possible that this is due to non-genic repetitive sequences, so we tried to align the sequence for one of these loci to the rest of the human genome. We selected one of these loci (NHEJ 1) which is very far from any gene (>20kb), as these regions are more prone to repetitive sequences, and tried to align 40bp fragments of a 1kb region around this locus, however we found no alternative alignments in the human genome.

In addition, it is unlikely in this mechanism that the depletion of POLD and POLE would yield different mutation signatures. Yes, POLE depletion could reduce this mutagenesis, but it is unlikely that POLD depletion would increase the same mutations while decreasing others. The differences we see between POLD and POLE are between different base substitutions, not between different loci. Overall, after this analysis, we do not believe that this mechanism is likely based on our data.

8. The analysis of BRCA2 and FANCA to RAD51 loading, while of interest, is not really to do with the mutations associated with DSB repair and feels like it should be part of a separate study.

We agree that there is a distinct difference in these experiments compared to those earlier in the manuscript, but given the data is likely of interest to the reader of this manuscript we would prefer to keep these results in the manuscript.

9. In contrast, the role of BRCA2 in resection, which is perhaps more related to the manuscript focus is not followed up.

This phenotype could lead to an interesting investigation, however given the length of the manuscript and that this data is not directly linked to the main message we are trying to convey, we feel additional experiments to study this our outside of the scope of this study. Especially since we believe the experiments required to investigate this would be very peripheral to those already completed and would require a lot of focus in the manuscript. We have expanded our interpretation of this data in the last paragraph of the results section (page 14, paragraph 4) to help integrate and expand on these findings.

Specific points:

P9: line 186: doesz to does.

P12: 264: knockdown of FANCA

P12: 266: similar to the RAD51 profile.

Thank you for all of these specific points, we have now corrected them within the text.

Reviewer #3 (Remarks to the Author):

Summary:

In this study the authors introduce a new technique for analysing the mutations arising at DSBs generated in the DiVA system developed by Gaelle Legube, a system involving the induction of DSBs at a subset of AsiS1 sites in the genome of U2OS cells (which should be discussed in the introduction with respect to its strengths and weaknesses). Bader et al perform NGS at such depth that they can catalogue mutations that occur during repair of this subset of sites. Importantly, their study exposes the mutagenic nature of repair by homologous recombination. HR is often considered “error-free” and the work of Bader and Bushell challenges this assumption, at least for HR-dependent repair at a subset of AsiS1 sites. They should point out that HR-dependent repair at other DSBs (e.g. DSBs generated during S phase) might not be so mutagenic. In fact, the persistent ‘cut and repair’ cycle likely to occur at each cut site until the enzyme is removed may select for mutations. Thus, the authors should not overinterpret their interesting results with the DiVA. In addition, their study supports previous data (which should be cited) that HR might function primarily to reduce translocations rather than mutations.

Major Points:

1. The introduction is admirably succinct and discusses the shortcomings of many technological approaches to DSB repair. However, although DiVA system developed by the Legube laboratory many years ago has been very useful to the field, they should also present the clear shortcomings of this system in their introduction.

This is very important thank you for pointing it out, a clear discussion of the disadvantages of the system, and our approaches here as a whole, has now been added (page 15 paragraph 2).

2. The chromatin environment is a critical consideration with respect to the AsiSI-induced DSBs in the DiVA system. They are believed to occur in euchromatic/transcribed regions, while AsiS1 sites that remain uncut are predominantly located in heterochromatic or poorly transcribed regions. The authors should include much more discussion, as well as interpretation, of the chromatin environment of the breaks, as well as speculation about how their results may extrapolate to DSBs at other genomic locations (e.g. heterochromatin, centromeric, telomeres, repetitive regions, fragile sites, rDNA, etc).

You are correct that the AsiSI loci that are cut skew towards open chromatin/transcribed regions, however there are still many loci that are in non-transcribed regions. In our design of the primer pool, we ensured a mix of low and high transcriptional activity loci for the uncut, NHEJ and HR prone regions to prevent bias in our results as a result of genomic features. We realise that we had not included a description of the design of the primer pool in the manuscript, we have now added this (page 4, paragraph 3) and have added the activity classification to Supplemental Table 1 that contains the information about the primers.

We also agree that even with this spread of transcriptional activity, our results are still at a specific subset of the genome as you say, excluding regions such as rDNA, telomeres etc. We have included this important point in our discussion of the limitations of iMUT-seq (page 15, paragraph 2).

3. Figure 3A. The “NHEJ” and “HR” categorisation needs to be revised because MRE11 and ATM/ATR regulate both these pathways. Remove these factors from the “HR progression” wedge and include in their own category at the beginning of the plot. Also, 53BP1 function upstream of Artemis, and RAD52 function in SSA not gene conversion and if included should be placed at the end of the plot (recently reviewed in Kieffer and Lowndes 2022).

We agree that RAD52 is distinctly SSA and this is also shown in our data, we have moved RAD52 to the end of the plots in Fig. 3 and it is no longer in the “HR progression” wedge.

Regarding the roles of MRN and ATM in NHEJ, this is a very good point and we agree with the points made in Kieffer and Lowndes 2022. The rigid use of the NHEJ and HR pathways is currently limiting since there is abundant evidence that these processes are more complex and overlapping than these classifications allow. However we also believe that models such as MRN promoting slow, resection-dependent NHEJ repair are still too controversial within the field to be used over the classically accepted repair pathways and are not as well characterised. But equally we agree that changes to this mode of thinking do need to be introduced in published literature, so we have adjusted the text to refer to these early HR factors as “resection promoting” rather than HR (page 12, paragraph 1). We have also added a paragraph to our discussion (page 16 paragraph 3) that talks about the models proposed in Kieffer and Lowndes 2022, and its cited literature, and how this intersects with our data, as we believe there are some aspects of our data that support these models. We hope that this will broaden readers understanding of these repair mechanisms and aid the adoption of these more appropriate and current models.

With the discrepancy of Artemis and 53BP1 we described Artemis as upstream of 53BP1 as 53BP1 appears to operate somewhat distinctly from the core NHEJ complex by antagonizing HR/resection alongside the Shieldin complex, whereas Artemis is directly involved NHEJ, albeit likely at a subset of breaks. In addition, we believe the data cited for Artemis being downstream of 53BP1 is more complex, as it does not show loss of Artemis recruitment with 53BP1 KO, but instead reduced retention at breaks over time⁽³⁾. Similar studies over shorter timeframes show Artemis recruiting within seconds to breaks in a similar manner to KU70/80⁽⁴⁾, consistent with a role in the Early response, whereas 53BP1 and BRCA1 are the Late response. We think this implicates Artemis as potentially prior to 53BP1, but with 53BP1 regulating Artemis retention, although since 53BP1 is more anti-HR/resection than it is pro-NHEJ it is possible these mechanisms are happening somewhat simultaneously.

4. Also, please comment on why the authors have used chemical inhibition instead of siRNA depletion for some of the proteins (e.g. in Figure 3A)? It would have been better to use siRNA depletion for all, this is especially important as pharmacological inhibition of PIK kinases may ‘lock’ these kinases onto DSBs thereby preventing accurate DSB repair. This is a quite different scenario than depleting PIK kinases and no doubt explains their result with DNA-PKi which, confusingly, in highly elevate mutation rate. It would be better to remove the PIKK inhibitor data from the manuscript and, if not, replace it with depletion data.

Since depletions also have their complications and caveats, we chose to use some inhibitors to provide an alternative to the siRNA that would hopefully also provide differential results important for understanding the effects of these drugs.

You are correct that the DNA-PKi results are explained by the locking of DNA-PK onto the DNA and we have discussed this in (page 9 paragraph 2). We still believe this is an interesting result, especially since this inhibitor is commonly used in molecular biology and is currently in pre-clinical studies for chemotherapy, it's mutagenic impacts are therefore quite important to understand.

As for ATM/ATR inhibition, they vary somewhat from the MRE11/BRCA1 siRNA depletions, but overall the result is very similar and we feel this is an effective orthogonal validation of these siRNA depletions.

We are happy to remove the DNA-PKi results as this would make the data more clear and easy to interpret, but would prefer to keep it if the reviewer agrees, as this would be beneficial data to present.

5. The use HCT116 cells in Figure 4F-H and 5H-I) is a bad choice as these MIN cells are known to have elevated levels of HR. The authors should repeat this analysis with more normal cells, e.g. RPE1 cells might be suitable. In this case, the authors should also consider using ICRF193 instead etoposide because the former is believed to result in a higher proportion of breaks being repaired by NHEJ.

We have now repeated the metaphase spread experiments for XRCC4/LIG4 depletion, since we believe that the depletion of KU70 causing genome instability is unlikely to vary as it is relatively well studied. We completed these new experiments using RPE-1 cells P53^{-/-} and using IR instead of etoposide (which should be highly NHEJ prone) and have added these results to the manuscript (Fig. 5h-i), moving our previous HCT116 results to the supplemental (Supplemental Fig. 5e-f). Interestingly, we see a stronger effect for LIG4 depletion in this experiment which we believe is likely due to the cell line and treatment differences you pointed out, so thank you for these suggestions.

6. Figure 6. It is inappropriate to use MRE11 depletion and ATM/Ri as readout for HR (see above). Retitle this figure and amend text accordingly.

We have altered the text to refer to these factors as "early resection factors" rather than HR. With regards to the title and other parts of this results section, the term HR is used as this is in reference to data also involving the polymerases (Fig. 6d-f) and now also our data from the TLR-assay (Supplemental Fig. 6d-h). Therefore we feel the term HR is appropriate in these contexts.

7. Figure 7. It is not surprising that RAD52 differs from BRCA2 and FANCA as it is primarily an SSA factor (see above). Retitle this figure and amend text accordingly.

We have now ensured that all references to RAD52 are as an SSA factor.

8. Include in Discussion text addressing why the analyses were limited to just 100bp either side of the AsiS1 sites as this 'distance' is still proximal given that resection can occur over much greater 'distances'.

This should definitely have been made clear previously and has now been added (page 4 paragraph 3/page 5 paragraph 1).

9. Include in Discussion text on the possibility of being able to perform similar analyses on other known genomic sites (e.g. G4s, sites of transcription/replication clashes, very highly expressed genes) upon further improvements to the required technology.

This is something we have given thought to in the past and would be good to have included. This is now in the discussion (page 15 paragraph 2).

Conclusion:

All in all, this study is well structured, thought-through adds provides new insight into DSB repair in human cells. However, the authors need to be careful to avoid over-interpretation. Furthermore, they should discuss in detail the limitations of the system used, the nature of the chromatin environment at the breaks and speculate on the likely mutagenic outcome at 'distance' further than 100bp away from the break (up to 3.5 Kb, Zhou et al (2013) NAR, Paull lab).

Minor points:

1. Supplemental Figure 2 legend is very confusing. If H is the same as E which is the same as C, would then H not be the same as C? Please clarify. Also state reasoning for using different types of statistics in different parts of Figure. Similarly for other Figure legends.

We have adjusted the figure legend to be more clear, we agree this was confusing. To make our use of statistical tests more clear, we have added a methods section on the statistical tests we used as they are relatively consistent throughout the manuscript.

2. Line 143/144. The authors claim that NHEJ induces large deletions around the break, where are the large deletions demonstrated and please define what is meant by 'large' in this case?

Our description of the results here was a bit vague, so we have re-written these lines to be more clear and referential to the data.

3. Use the word 'depletion' instead of 'knockdown' throughout as 'knockdown' is technical jargon.

We have changed all uses of knockdown to depletion.

4. Figure 4A in graph legend (coloured squares) is missing

We have added these in

5. Line 186 "does" is spelled incorrectly

We have now corrected this, thank you.

6. For microscopy images throughout grey scale images give better contrast and may well better

present the data.

We agree this would improve the contrast of the images and have changed all fluorescent images to greyscale.

7. Clearly state sequencing depths in main text.

This is an important point as the sensitivity of iMUT-seq is linked to the sequencing depth and it is therefore important we make this clear to the reader. We have added this into our description of the technique (page 5, paragraph 1).

REFERENCES

- (1) Aymard F, Aguirrebengoa M, Guillou E, Javierre BM, Bugler B, Arnould C, et al. Genome-wide mapping of long-range contacts unveils clustering of DNA double-strand breaks at damaged active genes. *Nature Structural & Molecular Biology* 2017;24(4):353-361.
- (2) Arnould C, Rocher V, Bader AS, Lesage E, Puget N, Clouaire T, et al. ATM-dependent formation of a novel chromatin compartment regulates the Response to DNA Double Strand Breaks and the biogenesis of translocations. *bioRxiv* 2021:2021.11.07.467654.
- (3) Wang J, Aroumougame A, Loblrich M, Li Y, Chen D, Chen J, et al. PTIP associates with Artemis to dictate DNA repair pathway choice. *Genes Dev* 2014 Dec 15;28(24):2693-2698.
- (4) Miller KM, Tjeertes JV, Coates J, Legube G, Polo SE, Britton S, et al. Human HDAC1 and HDAC2 function in the DNA-damage response to promote DNA nonhomologous end-joining. *Nature Structural & Molecular Biology* 2010;17(9):1144-1151.

REVIEWERS' COMMENTS

Reviewer #1 (Remarks to the Author):

In their revised manuscript, the authors have addressed the comments that were raised. The authors have provided an indepth response, and I am pleased to see the increased nuance in the manuscript, including in the discussion section. I have two points that could be further clarified.

Concerning the orthogonal approach to induce DNA breaks and measure mutagenic spectrs of HR, the authors now include data using the Trafficlight reporter. The obtained results confirm the mutagenic effects of HR repair, and these data make the manuscript much stronger. What it unclear to me, is how the authors technically subdivide the repair events into HR- and HR+. The methods section on this experiment is very limited. Also, it would have been good to include a flow cytometry plot of the cells upon Icel induction to demonstrate efficacy of break induction based on fluorescence switching.

Concerning the cell cycle comment I made earlier: I am not aware of a cell cycle synchronization protocol that does not potentially interfere with DNA integrity or repair kinetics. I was just wondering whether the AsiSI induction would induce a strong cell cycle arrest, which would be good to keep in mind when interpreting results.

Reviewer #2 (Remarks to the Author):

The manuscript is considerably improved and I appreciate the care and attention the authors have taken to address the various points raised. However, I still take issue with the claim in the abstract and elsewhere that 'Notably we found that homologous recombination is highly mutagenic. When challenged on this originally the authors accept that 'Our estimate for mutations per DSB is ~1.0-1.5% (page 7 paragraph 3) which we agree still suggests DSB repair is in general quite efficient and not as mutagenic as one might expect'.

Given that the authors claim of HR as highly mutagenic being based on the RELATIVE frequencies of HR induced mutations to the total number of mutations they observe, can I suggest that they simply change this to 'relatively mutagenic'?

This is an important point as I believe that making a claim that the HR pathway is highly mutagenic when it is clearly ~99% accurate is misrepresenting their otherwise careful and very well-presented study and has the potential to mislead the field. I do not believe that acceding this point diminishes the impact of the manuscript.

Reviewer #3 (Remarks to the Author):

The authors have substantially addressed all the issues raised in my review. Additionally, as this is of interests to specialists in the field, I am happy for the DNA-PKi data to be kept in the manuscript as requested by the authors.

The textual and experimental changes/additions to the manuscript have further strengthened this interesting study.

I strongly agree with the authors comments to reviewer 2 that "NHEJ is not as mutagenic", while "HR appears to be more mutagenic" than previously thought and agree with their arguments present in the discussion (p18). The field should move away from simplistic descriptions of HR and NHEJ as being 'error free' or 'error prone'.

REVIEWERS' COMMENTS

Reviewer #1 (Remarks to the Author):

In their revised manuscript, the authors have addressed the comments that were raised. The authors have provided an indepth response, and I am pleased to see the increased nuance in the manuscript, including in the discussion section. I have two points that could be further clarified.

Concerning the orthogonal approach to induce DNA breaks and measure mutagenic spectrs of HR, the authors now include data using the Traffilight reporter. The obtained results confirm the mutagenic effects of HR repair, and these data make the manuscript much stronger. What it unclear to me, is how the authors technically subdivide the repair events into HR- and HR+. The methods section on this experiment is very limited. Also, it would have been good to include a flow cytometry plot of the cells upon Iscel induction to demonstrate efficacy of break induction based on fluorescence switching.

Concerning the cell cycle comment I made earlier: I am not aware of a cell cycle synchronization protocol that does not potentially interfere with DNA integrity or repair kinetics. I was just wondering whether the AsiSI induction would induce a strong cell cycle arrest, which would be good to keep in mind when interpreting results.

We have added further detail to both the manuscript results and methods to make the process more clear (page 12, paragraph 4 and page 25, paragraph 1).

We had thought to include flow cytometry plots, however we felt it would be out of place, since no TLR flow cytometry experiments were carried out. We did some optimisation experiments using flow, but we chose to use bulk cell cultures for our NGS experiments as the sequencing would identify HR positive and negative reads without the need for FACS.

Reviewer #2 (Remarks to the Author):

The manuscript is considerably improved and I appreciate the care and attention the authors have taken to address the various points raised. However, I still take issue with the claim in the abstract and elsewhere that 'Notably we found that homologous recombination is highly mutagenic. When challenged on this originally the authors accept that 'Our estimate for mutations per DSB is ~1.0-1.5% (page 7 paragraph 3) which we agree still suggests DSB repair is in general quite efficient and not as mutagenic as one might expect'.

Given that the authors claim of HR as highly mutagenic being based on the RELATIVE frequencies of HR induced mutations to the total number of mutations they observe, can I suggest that they simply change this to 'relatively mutagenic'?

This is an important point as I believe that making a claim that the HR pathway is highly mutagenic when it is clearly ~99% accurate is misrepresenting their otherwise careful and very well-presented study and has the potential to mislead the field. I do not believe that acceding this point diminishes the impact of the manuscript.

We completely agree that our terminology should be shifted to relative rather than absolute terms. We see that the phrase 'highly mutagenic' can actually be misleading, especially

when read from a more general perspective. We have changed this sentence in the abstract and have gone through the manuscript to ensure that only relative terminology is used.

Reviewer #3 (Remarks to the Author):

The authors have substantially addressed all the issues raised in my review. Additionally, as this is of interests to specialists in the field, I am happy for the DNA-PKi data to be kept in the manuscript as requested by the authors.

The textual and experimental changes/additions to the manuscript have further strengthened this interesting study.

I strongly agree with the authors comments to reviewer 2 that "NHEJ is not as mutagenic", while "HR appears to be more mutagenic" than previously thought and agree with their arguments present in the discussion (p18). The field should move away from simplistic descriptions of HR and NHEJ as being 'error free' or 'error prone'.

We thank the reviewer for their comments and we have adjusted our manuscript in agreement with reviewer 2's comment.